# Rounding-based Moves for Metric Labeling

**M. Pawan Kumar**
Ecole Centrale Paris & INRIA Saclay
pawan.kumar@ecp.fr

## Abstract

Metric labeling is a special case of energy minimization for pairwise Markov random fields. The energy function consists of arbitrary unary potentials, and pairwise potentials that are proportional to a given metric distance function over the label set. Popular methods for solving metric labeling include (i) move-making algorithms, which iteratively solve a minimum $st$-cut problem; and (ii) the linear programming (LP) relaxation based approach. In order to convert the fractional solution of the LP relaxation to an integer solution, several randomized rounding procedures have been developed in the literature. We consider a large class of parallel rounding procedures, and design move-making algorithms that closely mimic them. We prove that the multiplicative bound of a move-making algorithm exactly matches the approximation factor of the corresponding rounding procedure for any arbitrary distance function. Our analysis includes all known results for move-making algorithms as special cases.

## 1 Introduction

A Markov random field (MRF) is a graph whose vertices are random variables, and whose edges specify a neighborhood over the random variables. Each random variable can be assigned a value from a set of labels, resulting in a labeling of the MRF. The putative labelings of an MRF are quantitatively distinguished from each other by an energy function, which is the sum of potential functions that depend on the cliques of the graph. An important optimization problem associate with the MRF framework is energy minimization, that is, finding a labeling with the minimum energy.

Metric labeling is a special case of energy minimization, which models several useful low-level vision tasks [3, 4, 18]. It is characterized by a finite, discrete label set and a metric distance function over the labels. The energy function in metric labeling consists of arbitrary unary potentials and pairwise potentials that are proportional to the distance between the labels assigned to them. The problem is known to be NP-hard [20]. Two popular approaches for metric labeling are: (i) move-making algorithms [4, 8, 14, 15, 21], which iteratively improve the labeling by solving a minimum $st$-cut problem; and (ii) linear programming (LP) relaxation [5, 13, 17, 22], which is obtained by dropping the integral constraints in the corresponding integer programming formulation. Move-making algorithms are very efficient due to the availability of fast minimum $st$-cut solvers [2] and are very popular in the computer vision community. In contrast, the LP relaxation is significantly slower, despite the development of specialized solvers [7, 9, 11, 12, 16, 19, 22, 23, 24, 25]. However, when used in conjunction with randomized rounding algorithms, the LP relaxation provides the best known polynomial-time theoretical guarantees for metric labeling [1, 5, 10].

At first sight, the difference between move-making algorithms and the LP relaxation appears to be the standard accuracy vs. speed trade-off. However, for some special cases of distance functions, it has been shown that appropriately designed move-making algorithms can match the theoretical guarantees of the LP relaxation [14, 15, 20]. In this paper, we extend this result for a large class of randomized rounding procedures, which we call parallel rounding. In particular we prove that for any arbitrary (semi-)metric distance function, there exist move-making algorithms that match the theoretical guarantees provided by parallel rounding. The proofs, the various corollaries of our

theorems (which cover all previously known guarantees) and our experimental results are deferred to the accompanying technical report.

## 2 Preliminaries

**Metric Labeling.** The problem of metric labeling is defined over an undirected graph $\mathbf{G} = (\mathbf{X}, \mathbf{E})$. The vertices $\mathbf{X} = \{X_1, X_2, \cdots, X_n\}$ are random variables, and the edges $\mathbf{E}$ specify a neighborhood relationship over the random variables. Each random variable can be assigned a value from the label set $\mathbf{L} = \{l_1, l_2, \cdots, l_h\}$. We assume that we are also provided with a metric distance function $d : \mathbf{L} \times \mathbf{L} \to \mathbb{R}^+$ over the labels.

We refer to an assignment of values to all the random variables as a labeling. In other words, a labeling is a vector $\mathbf{x} \in \mathbf{L}^n$, which specifies the label $x_a$ assigned to each random variable $X_a$. The $h^n$ different labelings are quantitatively distinguished from each other by an energy function $Q(\mathbf{x})$, which is defined as follows:

$$Q(\mathbf{x}) = \sum_{X_a \in \mathbf{X}} \theta_a(x_a) + \sum_{(X_a, X_b) \in \mathbf{E}} w_{ab} d(x_a, x_b).$$

Here, the unary potentials $\theta_a(\cdot)$ are arbitrary, and the edge weights $w_{ab}$ are non-negative. Metric labeling requires us to find a labeling with the minimum energy. It is known to be NP-hard.

**Multiplicative Bound.** As metric labeling plays a central role in low-level vision, several approximate algorithms have been proposed in the literature. A common theoretical measure of accuracy for an approximate algorithm is the multiplicative bound. In this work, we are interested in the multiplicative bound of an algorithm with respect to a distance function. Formally, given a distance function $d$, the multiplicative bound of an algorithm is said to be $B$ if the following condition is satisfied for all possible values of unary potentials $\theta_a(\cdot)$ and non-negative edge weights $w_{ab}$:

$$\sum_{X_a \in \mathbf{X}} \theta_a(\hat{x}_a) + \sum_{(X_a, X_b) \in \mathbf{E}} w_{ab} d(\hat{x}_a, \hat{x}_b) \le \sum_{X_a \in \mathbf{X}} \theta_a(x_a^*) + B \sum_{(X_a, X_b) \in \mathbf{E}} w_{ab} d(x_a^*, x_b^*). \quad (1)$$

Here, $\hat{\mathbf{x}}$ is the labeling estimated by the algorithm for the given values of unary potentials and edge weights, and $\mathbf{x}^*$ is an optimal labeling. Multiplicative bounds are greater than or equal to 1, and are invariant to reparameterizations of the unary potentials. A multiplicative bound $B$ is said to be tight if the above inequality holds as an equality for some value of unary potentials and edge weights.

**Linear Programming Relaxation.** An overcomplete representation of a labeling can be specified using the following variables: (i) unary variables $y_a(i) \in \{0, 1\}$ for all $X_a \in \mathbf{X}$ and $l_i \in \mathbf{L}$ such that $y_a(i) = 1$ if and only if $X_a$ is assigned the label $l_i$; and (ii) pairwise variables $y_{ab}(i, j) \in \{0, 1\}$ for all $(X_a, X_b) \in \mathbf{E}$ and $l_i, l_j \in \mathbf{L}$ such that $y_{ab}(i, j) = 1$ if and only if $X_a$ and $X_b$ are assigned labels $l_i$ and $l_j$ respectively. This allows us to formulate metric labeling as follows:

$$\min_{\mathbf{y}} \quad \sum_{X_a \in \mathbf{X}} \sum_{l_i \in \mathbf{L}} \theta_a(l_i) y_a(i) + \sum_{(X_a, X_b) \in \mathbf{E}} \sum_{l_i, l_j \in \mathbf{L}} w_{ab} d(l_i, l_j) y_{ab}(i, j),$$

$$\text{s.t.} \quad \sum_{l_i \in \mathbf{L}} y_a(i) = 1, \forall X_a \in \mathbf{X},$$

$$\sum_{l_j \in \mathbf{L}} y_{ab}(i, j) = y_a(i), \forall (X_a, X_b) \in \mathbf{E}, l_i \in \mathbf{L},$$

$$\sum_{l_i \in \mathbf{L}} y_{ab}(i, j) = y_b(j), \forall (X_a, X_b) \in \mathbf{E}, l_j \in \mathbf{L},$$

$$y_a(i) \in \{0, 1\}, y_{ab}(i, j) \in \{0, 1\}, \forall X_a \in \mathbf{X}, (X_a, X_b) \in \mathbf{E}, l_i, l_j \in \mathbf{L}.$$

By relaxing the final set of constraints such that the optimization variables can take any value between 0 and 1 inclusive, we obtain a linear program (LP). The computational complexity of solving the LP relaxation is polynomial in the size of the problem.

**Rounding Procedure.** In order to prove theoretical guarantees of the LP relaxation, it is common to use a rounding procedure that can covert a feasible fractional solution $\mathbf{y}$ of the LP relaxation to a feasible integer solution $\hat{\mathbf{y}}$ of the integer linear program. Several rounding procedures have been

proposed in the literature. In this work, we focus on the randomized parallel rounding procedures proposed in [5, 10]. These procedures have the property that, given a fractional solution $\mathbf{y}$, the probability of assigning a label $l_i \in \mathbf{L}$ to a random variable $X_a \in \mathbf{X}$ is equal to $y_a(i)$, that is,

$$\Pr(\hat{y}_a(i) = 1) = y_a(i). \tag{2}$$

We will describe the various rounding procedures in detail in sections 3-5. For now, we would like to note that our reason for focusing on the parallel rounding of [5, 10] is that they provide the best known polynomial-time theoretical guarantees for metric labeling. Specifically, we are interested in their approximation factor, which is defined next.

**Approximation Factor.** Given a distance function $d$, the approximation factor for a rounding procedure is said to be $F$ if the following condition is satisfied for all feasible fractional solutions $\mathbf{y}$:

$$\mathbb{E}\left( \sum_{l_i, l_j \in \mathbf{L}} d(l_i, l_j)\hat{y}_a(i)\hat{y}_b(j) \right) \leq F \sum_{l_i, l_j \in \mathbf{L}} d(l_i, l_j)y_{ab}(i, j). \tag{3}$$

Here, $\hat{\mathbf{y}}$ refers to the integer solution, and the expectation is taken with respect to the randomized rounding procedure applied to the feasible solution $\mathbf{y}$.

Given a rounding procedure with an approximation factor of $F$, an optimal fractional solution $\mathbf{y}^*$ of the LP relaxation can be rounded to a labeling $\hat{\mathbf{y}}$ that satisfies the following condition:

$$\mathbb{E}\left( \sum_{X_a \in \mathbf{X}} \sum_{l_i \in \mathbf{L}} \theta_a(l_i)\hat{y}_a(i) + \sum_{(X_a, X_b) \in \mathbf{E}} \sum_{l_i, l_j \in \mathbf{L}} w_{ab} d(l_i, l_j)\hat{y}_a(i)\hat{y}_b(j) \right)$$
$$\leq \sum_{X_a \in \mathbf{X}} \sum_{l_i \in \mathbf{L}} \theta_a(l_i)y_a^*(i) + F \sum_{(X_a, X_b) \in \mathbf{E}} \sum_{l_i, l_j \in \mathbf{L}} w_{ab} d(l_i, l_j)y_{ab}^*(i, j).$$

The above inequality follows directly from properties (2) and (3). Similar to multiplicative bounds, approximation factors are always greater than or equal to 1, and are invariant to reparameterizations of the unary potentials. An approximation factor $F$ is said to be tight if the above inequality holds as an equality for some value of unary potentials and edge weights.

**Submodular Energy Function.** We will use the following important fact throughout this paper. Given an energy function defined using arbitrary unary potentials, non-negative edge weights and a submodular distance function, an optimal labeling can be computed in polynomial time by solving an equivalent minimum $st$-cut problem [6]. Recall that a submodular distance function $d'$ over a label set $\mathbf{L} = \{l_1, l_2, \cdots, l_h\}$ satisfies the following properties: (i) $d'(l_i, l_j) \geq 0$ for all $l_i, l_j \in \mathbf{L}$, and $d'(l_i, l_j) = 0$ if and only if $i = j$; and (ii) $d'(l_i, l_j) + d'(l_{i+1}, l_{j+1}) \leq d'(l_i, l_{j+1}) + d'(l_{i+1}, l_j)$ for all $l_i, l_j \in \mathbf{L} \backslash \{l_h\}$ (where $\backslash$ refers to set difference).

## 3 Complete Rounding and Complete Move

We start with a simple rounding scheme, which we call complete rounding. While complete rounding is not very accurate, it would help illustrate the flavor of our results. We will subsequently consider its generalizations, which have been useful in obtaining the best-known approximation factors for various special cases of metric labeling.

The complete rounding procedure consists of a single stage where we use the set of all unary variables to obtain a labeling (as opposed to other rounding procedures discussed subsequently). Algorithm 1 describes its main steps. Intuitively, it treats the value of the unary variable $y_a(i)$ as the probability of assigning the label $l_i$ to the random variable $X_a$. It obtains a labeling by sampling from all the distributions $\mathbf{y}_a = [y_a(i), \forall l_i \in \mathbf{L}]$ simultaneously using the same random number.

It can be shown that using a different random number to sample the distributions $\mathbf{y}_a$ and $\mathbf{y}_b$ of two neighboring random variables $(X_a, X_b) \in \mathbf{E}$ results in an infinite approximation factor. For example, let $\overline{y}_a(i) = \overline{y}_b(i) = 1/h$ for all $l_i \in \mathbf{L}$, where $h$ is the number of labels. The pairwise variables $\overline{\mathbf{y}}_{ab}$ that minimize the energy function are $\overline{y}_{ab}(i, i) = 1/h$ and $\overline{y}_{ab}(i, j) = 0$ when $i \neq j$. For the above feasible solution of the LP relaxation, the RHS of inequality (3) is 0 for any finite $F$, while the LHS of inequality (3) is strictly greater than 0 if $h > 1$. However, we will shortly show that using the same random number $r$ for all random variables provides a finite approximation factor.

---
**Algorithm 1** *The complete rounding procedure.*
---
**input** A feasible solution **y** of the LP relaxation.
 1: Pick a real number $r$ uniformly from $[0, 1]$.
 2: **for all** $X_a \in \mathbf{X}$ **do**
 3:    Define $Y_a(0) = 0$ and $Y_a(i) = \sum_{j=1}^{i} y_a(j)$ for all $l_i \in \mathbf{L}$.
 4:    Assign the label $l_i \in \mathbf{L}$ to the random variable $X_a$ if $Y_a(i-1) < r \le Y_a(i)$.
 5: **end for**
---

We now turn our attention to designing a move-making algorithm whose multiplicative bound matches the approximation factor of the complete rounding procedure. To this end, we modify the range expansion algorithm proposed in [15] for truncated convex pairwise potentials to a general (semi-)metric distance function. Our method, which we refer to as the complete move-making algorithm, considers all putative labels of all random variables, and provides an approximate solution in a single iteration. Algorithm 2 describes its two main steps. First, it computes a submodular overestimation of the given distance function by solving the following optimization problem:

$$\overline{d} = \underset{d'}{\operatorname{argmin}} t \tag{4}$$
$$\text{s.t.} \quad d'(l_i, l_j) \le td(l_i, l_j), \forall l_i, l_j \in \mathbf{L},$$
$$d'(l_i, l_j) \ge d(l_i, l_j), \forall l_i, l_j \in \mathbf{L},$$
$$d'(l_i, l_j) + d'(l_{i+1}, l_{j+1}) \le d'(l_i, l_{j+1}) + d'(l_{i+1}, l_j), \forall l_i, l_j \in \mathbf{L} \backslash \{l_h\}.$$

The above problem minimizes the maximum ratio of the estimated distance to the original distance over all pairs of labels, that is, $\max_{i \ne j} d'(l_i, l_j)/d(l_i, l_j)$. We will refer to the optimal value of problem (4) as the submodular distortion of the distance function $d$. Second, it replaces the original distance function by the submodular overestimation and computes an approximate solution to the original metric labeling problem by solving a single minimum $st$-cut problem. Note that, unlike the range expansion algorithm [15] that uses the readily available submodular overestimation of a truncated convex distance (namely, the corresponding convex distance function), our approach estimates the submodular overestimation via the LP (4). Since the LP (4) can be solved for any arbitrary distance function, it makes complete move-making more generally applicable.

---
**Algorithm 2** *The complete move-making algorithm.*
---
**input** Unary potentials $\theta_a(\cdot)$, edge weights $w_{ab}$, distance function $d$.
 1: Compute a submodular overestimation of $d$ by solving problem (4).
 2: Using the approach of [6], solve the following problem via an equivalent minimum $st$-cut problem:
$$\hat{\mathbf{x}} = \underset{\mathbf{x} \in \mathbf{L}^n}{\operatorname{argmin}} \sum_{X_a \in \mathbf{X}} \theta_a(x_a) + \sum_{(X_a, X_b) \in \mathbf{E}} w_{ab}\overline{d}(x_a, x_b).$$
---

The following theorem establishes the theoretical guarantees of the complete move-making algorithm and the complete rounding procedure.

**Theorem 1.** *The tight multiplicative bound of the complete move-making algorithm is equal to the submodular distortion of the distance function. Furthermore, the tight approximation factor of the complete rounding procedure is also equal to the submodular distortion of the distance function.*

In terms of computational complexities, complete move-making is significantly faster than solving the LP relaxation. Specifically, given an MRF with $n$ random variables and $m$ edges, and a label set with $h$ labels, the LP relaxation requires at least $O(m^3 h^3 log(m^2 h^3))$ time, since it consists of $O(mh^2)$ optimization variables and $O(mh)$ constraints. In contrast, complete move-making requires $O(nmh^3 log(m))$ time, since the graph constructed using the method of [6] consists of $O(nh)$ nodes and $O(mh^2)$ arcs. Note that complete move-making also requires us to solve the linear program (4). However, since problem (4) is independent of the unary potentials and the edge weights, it only needs to be solved once beforehand in order to compute the approximate solution for any metric labeling problem defined using the distance function $d$.

## 4 Interval Rounding and Interval Moves

Theorem 1 implies that the approximation factor of the complete rounding procedure is very large for distance functions that are highly non-submodular. For example, consider the truncated linear distance function defined as follows over a label set $\mathbf{L} = \{l_1, l_2, \cdots, l_h\}$:

$$d(l_i, l_j) = \min\{|i - j|, M\}.$$

Here, $M$ is a user specified parameter that determines the maximum distance. The tightest submodular overestimation of the above distance function is the linear distance function, that is, $d(l_i, l_j) = |i - j|$. This implies that the submodular distortion of the truncated linear metric is $(h - 1)/M$, and therefore, the approximation factor for the complete rounding procedure is also $(h - 1)/M$. In order to avoid this large approximation factor, Chekuri *et al.* [5] proposed an interval rounding procedure, which captures the intuition that it is beneficial to assign similar labels to as many random variables as possible.

Algorithm 3 provides a description of interval rounding. The rounding procedure chooses an interval of at most $q$ consecutive labels (step 2). It generates a random number $r$ (step 3), and uses it to attempt to assign labels to previously unlabeled random variables from the selected interval (steps 4-7). It can be shown that the overall procedure converges in a polynomial number of iterations with a probability of 1 [5]. Note that if we fix $q = h$ and $z = 1$, interval rounding becomes equivalent to complete rounding. However, the analyses in [5, 10] shows that other values of $q$ provide better approximation factors for various special cases.

---

**Algorithm 3** *The interval rounding procedure.*

---

**input** A feasible solution $\mathbf{y}$ of the LP relaxation.

1: **repeat**
2:     Pick an integer $z$ uniformly from $[-q + 2, h]$. Define an interval of labels $\mathbf{I} = \{l_s, \cdots, l_e\}$, where $s = \max\{z, 1\}$ is the start index and $e = \min\{z + q - 1, h\}$ is the end index.
3:     Pick a real number $r$ uniformly from $[0, 1]$.
4:     **for all** Unlabeled random variables $X_a$ **do**
5:         Define $Y_a(0) = 0$ and $Y_a(i) = \sum_{j=s}^{s+i-1} y_a(j)$ for all $i \in \{1, \cdots, e - s + 1\}$.
6:         Assign the label $l_{s+i-1} \in \mathbf{I}$ to the $X_a$ if $Y_a(i - 1) < r \leq Y_a(i)$.
7:     **end for**
8: **until** All random variables have been assigned a label.

---

Our goal is to design a move-making algorithm whose multiplicative bound matches the approximation factor of interval rounding for any choice of $q$. To this end, we propose the interval move-making algorithm that generalizes the range expansion algorithm [15], originally proposed for truncated convex distances, to arbitrary distance functions. Algorithm 4 provides its main steps. The central idea of the method is to improve a given labeling $\hat{\mathbf{x}}$ by allowing each random variable $X_a$ to either retain its current label $\hat{x}_a$ or to choose a new label from an interval of consecutive labels. In more detail, let $\mathbf{I} = \{l_s, \cdots, l_e\} \subseteq \mathbf{L}$ be an interval of labels of length at most $q$ (step 4). For the sake of simplicity, let us assume that $\hat{x}_a \notin \mathbf{I}$ for any random variable $X_a$. We define $\mathbf{I}_a = \mathbf{I} \bigcup \{\hat{x}_a\}$ (step 5). For each pair of neighboring random variables $(X_a, X_b) \in \mathbf{E}$, we compute a submodular distance function $\overline{d}_{\hat{x}_a, \hat{x}_b} : \mathbf{I}_a \times \mathbf{I}_b \to \mathbb{R}^+$ by solving the following linear program (step 6):

$$\overline{d}_{\hat{x}_a, \hat{x}_b} = \underset{d'}{\operatorname{argmin}} \, t \tag{5}$$

$$\begin{aligned} \text{s.t.} \quad & d'(l_i, l_j) \leq t d(l_i, l_j), \forall l_i \in \mathbf{I}_a, l_j \in \mathbf{I}_b, \\ & d'(l_i, l_j) \geq d(l_i, l_j), \forall l_i \in \mathbf{I}_a, l_j \in \mathbf{I}_b, \\ & d'(l_i, l_j) + d'(l_{i+1}, l_{j+1}) \leq d'(l_i, l_{j+1}) + d'(l_{i+1}, l_j), \forall l_i, l_j \in \mathbf{I} \backslash \{l_e\}, \\ & d'(l_i, l_e) + d'(l_{i+1}, \hat{x}_b) \leq d'(l_i, \hat{x}_b) + d'(l_{i+1}, l_e), \forall l_i \in \mathbf{I} \backslash \{l_e\}, \\ & d'(l_e, l_j) + d'(\hat{x}_a, l_{j+1}) \leq d'(l_e, l_{j+1}) + d'(\hat{x}_a, l_j), \forall l_j \in \mathbf{I} \backslash \{l_e\}, \\ & d'(l_e, l_e) + d(\hat{x}_a, \hat{x}_b) \leq d'(l_e, \hat{x}_b) + d'(\hat{x}_a, l_e). \end{aligned}$$

Similar to problem (4), the above problem minimizes the maximum ratio of the estimated distance to the original distance. However, instead of introducing constraints for all pairs of labels, it is only

considers pairs of labels $l_i$ and $l_j$ where $l_i \in \mathbf{I}_a$ and $l_j \in \mathbf{I}_b$. Furthermore, it does not modify the distance between the current labels $\hat{x}_a$ and $\hat{x}_b$ (as can be seen in the last constraint of problem (5)).

Given the submodular distance functions $\overline{d}_{\hat{x}_a,\hat{x}_b}$, we can compute a new labeling $\overline{\mathbf{x}}$ by solving the following optimization problem via minimum $st$-cut using the method of [6] (step 7):

$$
\begin{aligned}
\overline{\mathbf{x}} = \quad & \underset{\mathbf{x}}{\operatorname{argmin}} \sum_{X_a \in \mathbf{X}} \theta_a(x_a) + \sum_{(X_a, X_b) \in \mathbf{E}} w_{ab} \overline{d}_{\hat{x}_a,\hat{x}_b}(x_a, x_b) \\
\text{s.t.} \quad & x_a \in \mathbf{I}_a, \forall X_a \in \mathbf{X}.
\end{aligned} \tag{6}
$$

If the energy of the new labeling $\overline{\mathbf{x}}$ is less than that of the current labeling $\hat{\mathbf{x}}$, then we update our labeling to $\overline{\mathbf{x}}$ (steps 8-10). Otherwise, we retain the current estimate of the labeling and consider another interval. The algorithm converges when the energy does not decrease for any interval of length at most $q$. Note that, once again, the main difference between interval move-making and the range expansion algorithm is the use of an appropriate optimization problem, namely the LP (5), to obtain a submodular overestimation of the given distance function. This allows us to use interval move-making for the general metric labeling problem, instead of focusing on only truncated convex models.

---

**Algorithm 4** *The interval move-making algorithm.*

---

**input** Unary potentials $\theta_a(\cdot)$, edge weights $w_{ab}$, distance function $d$, initial labeling $\mathbf{x}^0$.
 1: Set current labeling to initial labeling, that is, $\hat{\mathbf{x}} = \mathbf{x}^0$.
 2: **repeat**
 3:     **for all** $z \in [-q+2, h]$ **do**
 4:         Define an interval of labels $\mathbf{I} = \{l_s, \cdots, l_e\}$, where $s = \max\{z, 1\}$ is the start index and $e = \min\{z + q - 1, h\}$ is the end index.
 5:         Define $\mathbf{I}_a = \mathbf{I} \bigcup \{\hat{x}_a\}$ for all random variables $X_a \in \mathbf{X}$.
 6:         Obtain submodular overestimates $\overline{d}_{\hat{x}_a,\hat{x}_b}$ for each pair of neighboring random variables $(X_a, X_b) \in \mathbf{E}$ by solving problem (5).
 7:         Obtain a new labeling $\overline{\mathbf{x}}$ by solving problem (6).
 8:         **if** Energy of $\overline{\mathbf{x}}$ is less than energy of $\hat{\mathbf{x}}$ **then**
 9:             Update $\hat{\mathbf{x}} = \overline{\mathbf{x}}$.
10:         **end if**
11:     **end for**
12: **until** Energy cannot be decreased further.

---

The following theorem establishes the theoretical guarantees of the interval move-making algorithm and the interval rounding procedure.

**Theorem 2.** *The tight multiplicative bound of the interval move-making algorithm is equal to the tight approximation factor of the interval rounding procedure.*

An interval move-making algorithm that uses an interval length of $q$ runs for at most $O(h/q)$ iterations. This follows from a simple modification of the result by Gupta and Tardos [8] (specifically, theorem 3.7). Hence, the total time complexity of interval move-making is $O(nmhq^2 log(m))$, since each iteration solves a minimum $st$-cut problem of a graph with $O(nq)$ nodes and $O(mq^2)$ arcs. In other words, interval move-making is at most as computationally complex as complete move-making, which in turn is significantly less complex than solving the LP relaxation. Note that problem (5), which is required for interval move-making, is independent of the unary potentials and the edge weights. Hence, it only needs to be solved once beforehand for all pairs of labels $(\hat{x}_a, \hat{x}_b) \in \mathbf{L} \times \mathbf{L}$ in order to obtain a solution for any metric labeling problem defined using the distance function $d$.

## 5 Hierarchical Rounding and Hierarchical Moves

We now consider the most general form of parallel rounding that has been proposed in the literature, namely the hierarchical rounding procedure [10]. The rounding relies on a hierarchical clustering of the labels. Formally, we denote a hierarchical clustering of $m$ levels for the label set $\mathbf{L}$ by $\mathbf{C} = \{\mathbf{C}(i), i = 1, \cdots, m\}$. At each level $i$, the clustering $\mathbf{C}(i) = \{\mathbf{C}(i,j) \subseteq \mathbf{L}, j = 1, \cdots, h^i\}$ is

mutually exclusive and collectively exhaustive, that is,

$$\bigcup_j \mathbf{C}(i,j) = \mathbf{L}, \mathbf{C}(i,j) \cap \mathbf{C}(i,j') = \emptyset, \forall j \neq j'.$$

Furthermore, for each cluster $\mathbf{C}(i,j)$ at the level $i > 2$, there exists a unique cluster $\mathbf{C}(i-1,j')$ in the level $i - 1$ such that $\mathbf{C}(i,j) \subseteq \mathbf{C}(i-1,j')$. We call the cluster $\mathbf{C}(i-1,j')$ the parent of the cluster $\mathbf{C}(i,j)$ and define $p(i,j) = j'$. Similarly, we call $\mathbf{C}(i,j)$ a child of $\mathbf{C}(i-1,j')$. Without loss of generality, we assume that there exists a single cluster at level 1 that contains all the labels, and that each cluster at level $m$ contains a single label.

---

**Algorithm 5** *The hierarchical rounding procedure.*

---

**input** A feasible solution $\mathbf{y}$ of the LP relaxation.

1: Define $f_a^1 = 1$ for all $X_a \in \mathbf{X}$.
2: **for all** $i \in \{2, \cdots, m\}$ **do**
3:     **for all** $X_a \in \mathbf{X}$ **do**
4:         Define $z_a^i(j)$ for all $j \in \{1, \cdots, h^i\}$ as follows:

$$z_a^i(j) = \begin{cases} \sum_{k,l_k \in \mathbf{C}(i,j)} y_a(k) & \text{if } p(i,j) = f_a^{i-1}, \\ 0 & \text{otherwise.} \end{cases}$$

5:         Define $y_a^i(j)$ for all $j \in \{1, \cdots, h^i\}$ as follows:

$$y_a^i(j) = \frac{z_a^i(j)}{\sum_{j'=1}^{h^i} z_a^i(j')}$$

6:     **end for**
7:     Using a rounding procedure (complete or interval) on $\mathbf{y}^i = [y_a^i(j), \forall X_a \in \mathbf{X}, j \in \{1, \cdots, h^i\}]$, obtain an integer solution $\hat{\mathbf{y}}^i$.
8:     **for all** $X_a \in \mathbf{X}$ **do**
9:         Let $k_a \in \{1, \cdots, h^i\}$ such that $\hat{y}^i(k_a) = 1$. Define $f_a^i = k_a$.
10:     **end for**
11: **end for**
12: **for all** $X_a \in \mathbf{X}$ **do**
13:     Let $l_k$ be the unique label present in the cluster $\mathbf{C}(m, f_a^m)$. Assign $l_k$ to $X_a$.
14: **end for**

---

Algorithm 5 describes the hierarchical rounding procedure. Given a clustering $\mathbf{C}$, it proceeds in a top-down fashion through the hierarchy while assigning each random variable to a cluster in the current level. Let $f_a^i$ be the index of the cluster assigned to the random variable $X_a$ in the level $i$. In the first step, the rounding procedure assigns all the random variables to the unique cluster $\mathbf{C}(1,1)$ (step 1). At each step $i$, it assigns each random variable to a unique cluster in the level $i$ by computing a conditional probability distribution as follows. The conditional probability $y_a^i(j)$ of assigning the random variable $X_a$ to the cluster $\mathbf{C}(i,j)$ is proportional to $\sum_{l_k \in \mathbf{C}(i,j)} y_a(k)$ if $p(i,j) = f_a^{i-1}$ (steps 3-6). The conditional probability $y_a^i(j) = 0$ if $p(i,j) \neq f_a^{i-1}$, that is, a random variable cannot be assigned to a cluster $\mathbf{C}(i,j)$ if it wasn't assigned to its parent in the previous step. Using a rounding procedure (complete or interval) for $\mathbf{y}^i$, we obtain an assignment of random variables to the clusters at level $i$ (step 7). Once such an assignment is obtained, the values $f_a^i$ are computed for all random variables $X_a$ (steps 8-10). At the end of step $m$, hierarchical rounding would have assigned each random variable to a unique cluster in the level $m$. Since each cluster at level $m$ consists of a single label, this provides us with a labeling of the MRF (steps 12-14).

Our goal is to design a move-making algorithm whose multiplicative bound matches the approximation factor of the hierarchical rounding procedure for any choice of hierarchical clustering $\mathbf{C}$. To this end, we propose the hierarchical move-making algorithm, which extends the hierarchical graph cuts approach for hierarchically well-separated tree (HST) metrics proposed in [14]. Algorithm 6 provides its main steps. In contrast to hierarchical rounding, the move-making algorithm traverses the hierarchy in a bottom-up fashion while computing a labeling for each cluster in the current level. Let $\mathbf{x}^{i,j}$ be the labeling corresponding to the cluster $\mathbf{C}(i,j)$. At the first step, when considering the level $m$ of the clustering, all the random variables are assigned the same label. Specifically, $x_a^{m,j}$

**Algorithm 6** *The hierarchical move-making algorithm.*

---

**input** Unary potentials $\theta_a(\cdot)$, edge weights $w_{ab}$, distance function $d$.

1: **for all** $j \in \{1, \cdots, h\}$ **do**
2:     Let $l_k$ be the unique label is the cluster $\mathbf{C}(m, j)$. Define $x_a^{m,j} = l_k$ for all $X_a \in \mathbf{X}$.
3: **end for**
4: **for all** $i \in \{2, \cdots, m\}$ **do**
5:     **for all** $j \in \{1, \cdots, h^{m-i+1}\}$ **do**
6:         Define $\mathbf{L}_a^{m-i+1,j} = \{x_a^{m-i+2,j'}, p(m-i+2, j') = j, j' \in \{1, \cdots, h^{m-i+2}\}\}$.
7:         Using a move-making algorithm (complete or interval), compute the labeling $\mathbf{x}^{m-i+1,j}$ under the constraint $x_a^{m-i+1,j} \in \mathbf{L}_a^{m-i+1,j}$.
8:     **end for**
9: **end for**
10: The final solution is $\mathbf{x}^{1,1}$.

---

is equal to the unique label contained in the cluster $\mathbf{C}(m, j)$ (steps 1-3). At step $i$, it computes the labeling $\mathbf{x}^{m-i+1,j}$ for each cluster $\mathbf{C}(m - i + 1, j)$ by using the labelings computed in the previous step. Specifically, it restricts the label assigned to a random variable $X_a$ in the labeling $\mathbf{x}^{m-i+1,j}$ to the subset of labels that were assigned to it by the labelings corresponding to the children of $\mathbf{C}(m - i + 1, j)$ (step 6). Under this restriction, the labeling $\mathbf{x}^{m-i+1,j}$ is computed by approximately minimizing the energy using a move-making algorithm (step 7). Implicit in our description is the assumption that that we will use a move-making algorithm (complete or interval) in step 7 of Algorithm 6 whose multiplicative bound matches the approximation factor of the rounding procedure (complete or interval) used in step 7 of Algorithm 5. Note that, unlike the hierarchical graph cuts approach [14], the hierarchical move-making algorithm can be used for any arbitrary clustering and not just the one specified by an HST metric.

The following theorem establishes the theoretical guarantees of the hierarchical move-making algorithm and the hierarchical rounding procedure.

**Theorem 3.** *The tight multiplicative bound of the hierarchical move-making algorithm is equal to the tight approximation factor of the hierarchical rounding procedure.*

Note that hierarchical move-making solves a series of problems defined on a smaller label set. Since the complexity of complete and interval move-making is superlinear in the number of labels, it can be verified that the hierarchical move-making algorithm is at most as computationally complex as the complete move-making algorithm (corresponding to the case when the clustering consists of only one cluster that contains all the labels). Hence, hierarchical move-making is significantly faster than solving the LP relaxation.

## 6 Discussion

For any general distance function that can be used to specify the (semi-)metric labeling problem, we proved that the approximation factor of a large family of parallel rounding procedures is matched by the multiplicative bound of move-making algorithms. This generalizes previously known results on the guarantees of move-making algorithms in two ways: (i) in contrast to previous results [14, 15, 20] that focused on special cases of distance functions, our results are applicable to arbitrary semi-metric distance functions; and (ii) the guarantees provided by our theorems are tight. Our experiments (described in the technical report) confirm that the rounding-based move-making algorithms provide similar accuracy to the LP relaxation, while being significantly faster due to the use of efficient minimum $st$-cut solvers.

Several natural questions arise. What is the exact characterization of the rounding procedures for which it is possible to design matching move-making algorithms? Can we design rounding-based move-making algorithms for other combinatorial optimization problems? Answering these questions will not only expand our theoretical understanding, but also result in the development of efficient and accurate algorithms.

**Acknowledgements.** This work is funded by the European Research Council under the European Community's Seventh Framework Programme (FP7/2007-2013)/ERC Grant agreement number 259112.

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
