[Supplementary Material]

# Rounding-based Moves for Metric Labeling

**M. Pawan Kumar**
Ecole Centrale Paris & INRIA Saclay
`pawan.kumar@ecp.fr`

## Abstract

Metric labeling is a special case of energy minimization for pairwise Markov random fields. The energy function consists of arbitrary unary potentials, and pairwise potentials that are proportional to a given metric distance function over the label set. Popular methods for solving metric labeling include (i) move-making algorithms, which iteratively solve a minimum $st$-cut problem; and (ii) the linear programming (LP) relaxation based approach. In order to convert the fractional solution of the LP relaxation to an integer solution, several randomized rounding procedures have been developed in the literature. We consider a large class of parallel rounding procedures, and design move-making algorithms that closely mimic them. We prove that the multiplicative bound of a move-making algorithm exactly matches the approximation factor of the corresponding rounding procedure for any arbitrary distance function. Our analysis includes all known results for move-making algorithms as special cases.

## 1 Introduction

A Markov random field (MRF) is a graph whose vertices are random variables, and whose edges specify a neighborhood over the random variables. Each random variable can be assigned a value from a set of labels, resulting in a labeling of the MRF. The putative labelings of an MRF are quantitatively distinguished from each other by an energy function, which is the sum of potential functions that depend on the cliques of the graph. An important optimization problem associate with the MRF framework is energy minimization, that is, finding a labeling with the minimum energy.

Metric labeling is a special case of energy minimization, which models several useful low-level vision tasks [3, 4, 21]. It is characterized by a finite, discrete label set and a metric distance function over the labels. The energy function in metric labeling consists of arbitrary unary potentials and pairwise potentials that are proportional to the distance between the labels assigned to them. The problem is known to be NP-hard [23]. Two popular approaches for metric labeling are: (i) move-making algorithms [4, 9, 15, 16, 24], which iteratively improve the labeling by solving a minimum $st$-cut problem; and (ii) linear programming (LP) relaxation [5, 14, 20, 25], which is obtained by dropping the integral constraints in the corresponding integer programming formulation. Move-making algorithms are very efficient due to the availability of fast minimum $st$-cut solvers [2] and are very popular in the computer vision community. In contrast, the LP relaxation is significantly slower, despite the development of specialized solvers [8, 10, 12, 13, 19, 22, 25, 26, 27, 28]. However, when used in conjunction with randomized rounding algorithms, the LP relaxation provides the best known polynomial-time theoretical guarantees for metric labeling [1, 5, 11].

At first sight, the difference between move-making algorithms and the LP relaxation appears to be the standard accuracy vs. speed trade-off. However, for some special cases of distance functions, it has been shown that appropriately designed move-making algorithms can match the theoretical guarantees of the LP relaxation [15, 16, 23]. In this paper, we extend this result for a large class of randomized rounding procedures, which we call parallel rounding. In particular we prove that for any arbitrary (semi-)metric distance function, there exist move-making algorithms that match the theoretical guarantees provided by parallel rounding. Our proofs are constructive, which allows us to test the rounding-based move-making algorithms empirically. Our experimental results are along

the same lines as those that were previously reported for various special distance functions [15, 16]. Specifically, they confirm that rounding-based moves provide similar accuracy to the LP relaxation while being significantly faster.

## 2 Preliminaries

**Metric Labeling.** The problem of metric labeling is defined over an undirected graph $\mathbf{G} = (\mathbf{X}, \mathbf{E})$. The vertices $\mathbf{X} = \{X_1, X_2, \cdots, X_n\}$ are random variables, and the edges $\mathbf{E}$ specify a neighborhood relationship over the random variables. Each random variable can be assigned a value from the label set $\mathbf{L} = \{l_1, l_2, \cdots, l_h\}$. We assume that we are also provided with a metric distance function $d : \mathbf{L} \times \mathbf{L} \to \mathbb{R}^+$ over the labels. Recall that a metric distance function satisfies the following properties: (i) $d(l_i, l_j) \geq 0$ for all $l_i, l_j \in \mathbf{L}$, and $d(l_i, l_j) = 0$ if and only if $i = j$; and (ii) $d(l_i, l_j) + d(l_j, l_k) \geq d(l_i, l_k)$ for all $l_i, l_j, l_k \in \mathbf{L}$.

We refer to an assignment of values to all the random variables as a labeling. In other words, a labeling is a vector $\mathbf{x} \in \mathbf{L}^n$, which specifies the label $x_a$ assigned to each random variable $X_a$. The $h^n$ different labelings are quantitatively distinguished from each other by an energy function $Q(\mathbf{x})$, which is defined as follows:

$$Q(\mathbf{x}) = \sum_{X_a \in \mathbf{X}} \theta_a(x_a) + \sum_{(X_a, X_b) \in \mathbf{E}} w_{ab} d(x_a, x_b).$$

Here, the unary potentials $\theta_a(\cdot)$ are arbitrary, and the edge weights $w_{ab}$ are non-negative. Metric labeling requires us to find a labeling with the minimum energy. It is known to be NP-hard.

**Multiplicative Bound.** As metric labeling plays a central role in low-level vision, several approximate algorithms have been proposed in the literature. A common theoretical measure of accuracy for an approximate algorithm is the multiplicative bound. In this work, we are interested in the multiplicative bound of an algorithm with respect to a distance function. Formally, given a distance function $d$, the multiplicative bound of an algorithm is said to be $B$ if the following condition is satisfied for all possible values of unary potentials $\theta_a(\cdot)$ and non-negative edge weights $w_{ab}$:

$$\sum_{X_a \in \mathbf{X}} \theta_a(\hat{x}_a) + \sum_{(X_a, X_b) \in \mathbf{E}} w_{ab} d(\hat{x}_a, \hat{x}_b) \leq \sum_{X_a \in \mathbf{X}} \theta_a(x_a^*) + B \sum_{(X_a, X_b) \in \mathbf{E}} w_{ab} d(x_a^*, x_b^*). \quad (1)$$

Here, $\hat{\mathbf{x}}$ is the labeling estimated by the algorithm for the given values of unary potentials and edge weights, and $\mathbf{x}^*$ is an optimal labeling. Multiplicative bounds are greater than or equal to 1, and are invariant to reparameterizations of the unary potentials. A multiplicative bound $B$ is said to be tight if the above inequality holds as an equality for some value of unary potentials and edge weights.

**Linear Programming Relaxation.** An overcomplete representation of a labeling can be specified using the following variables: (i) unary variables $y_a(i) \in \{0, 1\}$ for all $X_a \in \mathbf{X}$ and $l_i \in \mathbf{L}$ such that $y_a(i) = 1$ if and only if $X_a$ is assigned the label $l_i$; and (ii) pairwise variables $y_{ab}(i, j) \in \{0, 1\}$ for all $(X_a, X_b) \in \mathbf{E}$ and $l_i, l_j \in \mathbf{L}$ such that $y_{ab}(i, j) = 1$ if and only if $X_a$ and $X_b$ are assigned labels $l_i$ and $l_j$ respectively. This allows us to formulate metric labeling as follows:

$$\min_{\mathbf{y}} \quad \sum_{X_a \in \mathbf{X}} \sum_{l_i \in \mathbf{L}} \theta_a(l_i) y_a(i) + \sum_{(X_a, X_b) \in \mathbf{E}} \sum_{l_i, l_j \in \mathbf{L}} w_{ab} d(l_i, l_j) y_{ab}(i, j),$$

$$\text{s.t.} \quad \sum_{l_i \in \mathbf{L}} y_a(i) = 1, \forall X_a \in \mathbf{X},$$

$$\sum_{l_j \in \mathbf{L}} y_{ab}(i, j) = y_a(i), \forall (X_a, X_b) \in \mathbf{E}, l_i \in \mathbf{L},$$

$$\sum_{l_i \in \mathbf{L}} y_{ab}(i, j) = y_b(j), \forall (X_a, X_b) \in \mathbf{E}, l_j \in \mathbf{L},$$

$$y_a(i) \in \{0, 1\}, y_{ab}(i, j) \in \{0, 1\}, \forall X_a \in \mathbf{X}, (X_a, X_b) \in \mathbf{E}, l_i, l_j \in \mathbf{L}.$$

The first set of constraints ensures that each random variables is assigned exactly one label. The second and third sets of constraints ensure that, for binary optimization variables, $y_{ab}(i, j) = y_a(i) y_b(j)$. By relaxing the final set of constraints such that the optimization variables can take any value between 0 and 1 inclusive, we obtain a linear program (LP). The computational complexity of solving the LP relaxation is polynomial in the size of the problem.

**Rounding Procedure.** In order to prove theoretical guarantees of the LP relaxation, it is common to use a rounding procedure that can covert a feasible fractional solution $\mathbf{y}$ of the LP relaxation to a feasible integer solution $\hat{\mathbf{y}}$ of the integer linear program. Several rounding procedures have been proposed in the literature. In this work, we focus on the randomized parallel rounding procedures proposed in [5, 11]. These procedures have the property that, given a fractional solution $\mathbf{y}$, the probability of assigning a label $l_i \in \mathbf{L}$ to a random variable $X_a \in \mathbf{X}$ is equal to $y_a(i)$, that is,

$$\Pr(\hat{y}_a(i) = 1) = y_a(i). \tag{2}$$

We will describe the various rounding procedures in detail in sections 3-5. For now, we would like to note that our reason for focusing on the parallel rounding of [5, 11] is that they provide the best known polynomial-time theoretical guarantees for metric labeling. Specifically, we are interested in their approximation factor, which is defined next.

**Approximation Factor.** Given a distance function $d$, the approximation factor for a rounding procedure is said to be $F$ if the following condition is satisfied for all feasible fractional solutions $\mathbf{y}$:

$$\mathbb{E}\left(\sum_{l_i, l_j \in \mathbf{L}} d(l_i, l_j)\hat{y}_a(i)\hat{y}_b(j)\right) \leq F \sum_{l_i, l_j \in \mathbf{L}} d(l_i, l_j)y_{ab}(i, j). \tag{3}$$

Here, $\hat{\mathbf{y}}$ refers to the integer solution, and the expectation is taken with respect to the randomized rounding procedure applied to the feasible solution $\mathbf{y}$.

Given a rounding procedure with an approximation factor of $F$, an optimal fractional solution $\mathbf{y}^*$ of the LP relaxation can be rounded to a labeling $\hat{\mathbf{y}}$ that satisfies the following condition:

$$\mathbb{E}\left(\sum_{X_a \in \mathbf{X}} \sum_{l_i \in \mathbf{L}} \theta_a(l_i)\hat{y}_a(i) + \sum_{(X_a, X_b) \in \mathbf{E}} \sum_{l_i, l_j \in \mathbf{L}} w_{ab}d(l_i, l_j)\hat{y}_a(i)\hat{y}_b(j)\right)$$
$$\leq \sum_{X_a \in \mathbf{X}} \sum_{l_i \in \mathbf{L}} \theta_a(l_i)y_a^*(i) + F \sum_{(X_a, X_b) \in \mathbf{E}} \sum_{l_i, l_j \in \mathbf{L}} w_{ab}d(l_i, l_j)y_{ab}^*(i, j).$$

The above inequality follows directly from properties (2) and (3). Similar to multiplicative bounds, approximation factors are always greater than or equal to 1, and are invariant to reparameterizations of the unary potentials. An approximation factor $F$ is said to be tight if the above inequality holds as an equality for some value of unary potentials and edge weights.

Approximation factors are closely linked to the integrality gap of the LP relaxation (roughly speaking, the ratio of the optimal value of the integer linear program to the optimal value of the relaxation), which in turn is related to the computational hardness of the metric labeling problem [17]. However, establishing the integrality gap of the LP relaxation for a given distance function is beyond the scope of this work. We are only interested in designing move-making algorithms whose multiplicative bounds match the approximation factors of the parallel rounding procedures.

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

The proof of theorem 2 is given in Appendix B. While Algorithms 3 and 4 use intervals of consecutive labels, they can easily be modified to use subsets of (potentially non-consecutive) labels. Our analysis could be extended to show that the multiplicative bound of the resulting subset move-making algorithm matches the approximation factor of the subset rounding procedure. However, our reason for focusing on intervals of consecutive labels is that several special cases of theorem 2 have previously been considered separately in the literature [9, 15, 16, 23]. Specifically, the following known results are corollaries of the above theorem. Note that, while the following corollaries have been previously proved in the literature, our work is the first to establish the tightness of the theoretical guarantees.

**Corollary 2.** *When $q = 1$, the multiplicative bound of the interval move-making algorithm (which is equivalent to the expansion algorithm) for the uniform metric distance is 2.*

The above corollary follows from the approximation factor of the interval rounding procedure proved in [11], but it was independently proved in [23].

**Corollary 3.** *When $q = M$, the multiplicative bound of the interval move-making algorithm for the truncated linear distance function is 4.*

The above corollary follows from the approximation factor of the interval rounding procedure proved in [5], but it was independently proved in [9].

**Corollary 4.** *When $q = \sqrt{2}M$, the multiplicative bound of the interval move-making algorithm for the truncated linear distance function is $2 + \sqrt{2}$.*

The above corollary follows from the approximation factor of the interval rounding procedure proved in [5], but it was independently proved in [16]. Finally, since our analysis does not use the triangular inequality of metric distance functions, it is also applicable to semi-metric labeling. Therefore, we can also state the following corollary for the truncated quadratic distance.

**Corollary 5.** *When $q = \sqrt{M}$, the multiplicative bound of the interval move-making algorithm for the truncated linear distance function is $O(\sqrt{M})$.*

The above corollary follows from the approximation factor of the interval rounding procedure proved in [5], but it was independently proved in [16].

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

The proof of the above theorem is given in Appendix C. The following known result is its corollary.

**Corollary 6.** *The multiplicative bound of the hierarchical move-making algorithm is $O(1)$ for an* HST *metric distance.*

The above corollary follows from the approximation factor of the hierarchical rounding procedure proved in [11], but it was independently proved in [15]. It is worth noting that the above result was also used to obtain an approximation factor of $O(\log h)$ for the general metric labeling problem in [11] and a matching multiplicative bound of $O(\log h)$ in [15].

Note that hierarchical move-making solves a series of problems defined on a smaller label set. Since the complexity of complete and interval move-making is superlinear in the number of labels, it can be verified that the hierarchical move-making algorithm is at most as computationally complex as the complete move-making algorithm (corresponding to the case when the clustering consists of only one cluster that contains all the labels). Hence, hierarchical move-making is significantly faster than solving the LP relaxation.

## 6 Experiments

We demonstrate the efficacy of rounding-based moves by comparing them to several state of the art methods using both synthetic and real data.

### 6.1 Synthetic Experiments

**Data.** We generated random grid MRFs of size $100 \times 100$, where each random variable can take one of 10 labels. The unary potentials were sampled from a uniform distribution over $[0, 10]$. The edge weights were sampled from a uniform distribution over $[0, 3]$. We considered four types of pairwise potentials: (i) truncated linear metric, where the truncation is sampled from a uniform distribution over $[1, 5]$; (ii) truncated quadratic metric, where the truncation is sampled from a uniform distribution over $[1, 25]$; (iii) random metrics, generated by computing the shortest path on graphs whose vertices correspond to the labels and whose edge lengths are uniformly distributed over $[1, 10]$; (iv) random semi-metrics, where the distance between two labels is sampled from a uniform distribution over $[1, 10]$. For each type of pairwise potentials, we generated 500 different MRFs.

**Methods.** We report results obtained by the following state of the art methods: (i) belief propagation (BP) [18]; (ii) sequential tree-reweighted message passing (TRW) [12], which optimizes the dual of the LP relaxation, and provides comparable results to other LP relaxation based approaches; (iii) expansion algorithm (EXP) [4]; and (iv) swap algorithm (SWAP) [3]. We compare the above

Figure 1: Results for the synthetic dataset. The x-axis shows the time in seconds, while the y-axis shows the energy value. The dashed line shows the value of the dual of the LP obtained by TRW. Best viewed in color.

methods to a hierarchy move-making algorithm (HIER), where a set of hierarchies is obtained by approximating a given (semi-)metric as a mixture of r-HST metrics using the method defined in [6]. We refer the reader to [6, 15] for details. Each subproblem of the hierarchical move-making algorithm is solved by interval move-making with interval length $q = 1$ (which corresponds to the expansion algorithm). In addition, for the truncated linear and truncated quadratic cases, we present results of interval move-making (INT) using the optimal interval length reported in [16].

**Results.**    Fig. 1 shows the results of the above methods. In terms of the energy, TRW is the most accurate. However, it is slow as it optimizes the dual of the LP relaxation. The labelings obtained by BP have high energy values. The standard move-making algorithms, EXP and SWAP, are fast due to the use of efficient minimum $st$-cut solvers. However, they are not as accurate as TRW. For the truncated linear and quadratic pairwise potentials, INT provides labelings with comparable energy to those of TRW, and is also computationally efficient. However, for general metrics and semi-metrics, it is not obvious how to obtain the optimal interval length. The HIER method is more generally applicable as there exist standard methods to approximate a (semi-)metric with a mixture of r-HST metrics [6]. It provides very accurate labelings (comparable to TRW), and is efficient in practice as it relies on solving each subproblem using an iterative move-making algorithm.

## 6.2    Dense Stereo Correspondence

**Data.**    Given two epipolar rectified images of the same scene, the problem of dense stereo correspondence requires us to obtain a correspondence between the pixels of the images. This problem

can be modeled as metric labeling, where the random variables represent the pixels of one of the images, and the labels represent the disparity values. A disparity label $l_i$ for a random variable $X_a$ representing a pixel $(u_a, v_a)$ of an image indicates that its corresponding pixel lies in location $(u_a + i, v_a)$. For the above problem, we use the unary potentials and edge weights that are specified in [21]. We use two types of pairwise potentials: (i) truncated linear with the truncation set at 4; and (ii) truncated quadratic with the truncation set at 16.

**Methods.** We report results on all the baseline methods that were used in the synthetic experiments, namely, BP, TRW, EXP, and SWAP. Since the pairwise potentials are either truncated linear or truncated quadratic, we report results for the interval move-making algorithm INT, which uses the optimal value of the interval length. We also show the results obtained by the hierarchical move-making algorithm (HIER), where once again the hierarchies are obtained by approximating the (semi-)metric as a mixture of r-HST metrics.

**Results.** Fig. 2-Fig. 7 shows the results for various standard pairs of images. Note that, similar to the synthetic experiments, TRW is the most accurate in terms of energy, but it is computationally inefficient. The results obtained by BP are not accurate. The standard move-making algorithms, EXP and SWAP, are fast but not as accurate as TRW. Among the rounding-based move-making algorithms INT is slower as it solves a minimum $st$-cut problem on a large graph at each iteration. In contrast, HIER uses an interval length of 1 for each subproblem and is therefore more efficient. The energy obtained by HIER is comparable to TRW.

| Image 1 | Image 2 | Ground Truth |

| BP | TRW | SWAP |
| Time=9.1s, Energy=686350 | Time=55.8s, Energy=654128 | Time=4.4s, Energy=668031 |

| EXP | INT | HIER |
| Time=3.3s, Energy=657005 | Time=87.2s, Energy=656945 | Time=34.6s, Energy=654557 |

Figure 2: Results for the 'tsukuba' image pair with truncated linear pairwise potentials.

|  |  |  |
|---|---|---|
| Image 1 | Image 2 | Ground Truth |
| BP<br>Time=29.9s, Energy=1586856 | TRW<br>Time=115.9s, Energy=1415343 | SWAP<br>Time=7.1s, Energy=1562459 |
| EXP<br>Time=5.1s, Energy=1546777 | INT<br>Time=275.6s, Energy=1533114 | HIER<br>Time=40.7s, Energy=1499134 |

Figure 3: Results for the 'tsukuba' image pair with truncated quadratic pairwise potentials.

## 7  Discussion

For any general distance function that can be used to specify the (semi-)metric labeling problem, we proved that the approximation factor of a large family of parallel rounding procedures is matched by the multiplicative bound of move-making algorithms. This generalizes previously known results on the guarantees of move-making algorithms in two ways: (i) in contrast to previous results [15, 16, 23] that focused on special cases of distance functions, our results are applicable to arbitrary semi-metric distance functions; and (ii) the guarantees provided by our theorems are tight. Our experiments confirm that the rounding-based move-making algorithms provide similar accuracy to the LP relaxation, while being significantly faster due to the use of efficient minimum $st$-cut solvers.

Several natural questions arise. What is the exact characterization of the rounding procedures for which it is possible to design matching move-making algorithms? Can we design rounding-based move-making algorithms for other combinatorial optimization problems? Answering these questions will not only expand our theoretical understanding, but also result in the development of efficient and accurate algorithms.

**Acknowledgements.**  This work is funded by the European Research Council under the European Community's Seventh Framework Programme (FP7/2007-2013)/ERC Grant agreement number 259112.

| Image 1 | Image 2 | Ground Truth |
| --- | --- | --- |

BP
Time=16.4s, Energy=3003629

TRW
Time=105.2s, Energy=2943481

SWAP
Time=7.7s, Energy=2954819

EXP
Time=11.5s, Energy=2953157

INT
Time=273.1s, Energy=2959133

HIER
Time=105.7s, Energy=2946177

Figure 4: Results for the 'venus' image pair with truncated linear pairwise potentials.

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

| Image 1 | Image 2 | Ground Truth |

BP
Time=54.3s, Energy=4183829

TRW
Time=223.0s, Energy=3080619

SWAP
Time=22.8s, Energy=3240891

EXP
Time=30.3s, Energy=3326685

INT
Time=522.3s, Energy=3216829

HIER
Time=113s, Energy=3210882

Figure 5: Results for the 'venus' image pair with truncated quadratic pairwise potentials.

[7] B. Flach and D. Schlesinger. Transforming an arbitrary minsum problem into a binary one. Technical report, TU Dresden, 2006.

[8] A. Globerson and T. Jaakkola. Fixing max-product: Convergent message passing algorithms for MAP LP-relaxations. In *NIPS*, 2007.

[9] A. Gupta and E. Tardos. A constant factor approximation algorithm for a class of classification problems. In *STOC*, 2000.

[10] T. Hazan and A. Shashua. Convergent message-passing algorithms for inference over general graphs with convex free energy. In *UAI*, 2008.

[11] J. Kleinberg and E. Tardos. Approximation algorithms for classification problems with pairwise relationships: Metric labeling and Markov random fields. In *STOC*, 1999.

[12] V. Kolmogorov. Convergent tree-reweighted message passing for energy minimization. *PAMI*, 2006.

[13] N. Komodakis, N. Paragios, and G. Tziritas. MRF optimization via dual decomposition: Message-passing revisited. In *ICCV*, 2007.

| Image 1 | Image 2 | Ground Truth |

| BP | TRW | SWAP |
| Time=47.5s, Energy=1771965 | Time=317.7s, Energy=1605057 | Time=35.2s, Energy=1606891 |

| EXP | INT | HIER |
| Time=26.5s, Energy=1603057 | Time=878.5s, Energy=1606558 | Time=313.7s, Energy=1596279 |

Figure 6: Results for the 'teddy' image pair with truncated linear pairwise potentials.

[14] A. Koster, C. van Hoesel, and A. Kolen. The partial constraint satisfaction problem: Facets and lifting theorems. *Operations Research Letters*, 1998.

[15] M. P. Kumar and D. Koller. MAP estimation of semi-metric MRFs via hierarchical graph cuts. In *UAI*, 2009.

[16] M. P. Kumar and P. Torr. Improved moves for truncated convex models. In *NIPS*, 2008.

[17] R. Manokaran, J. Naor, P. Raghavendra, and R. Schwartz. SDP gaps and UGC hardness for multiway cut, 0-extension and metric labeling. In *STOC*, 2008.

[18] J. Pearl. *Probabilistic Reasoning in Intelligent Systems: Networks of Plausible Inference*. Morgan Kauffman, 1998.

[19] P. Ravikumar, A. Agarwal, and M. Wainwright. Message-passing for graph-structured linear programs: Proximal projections, convergence and rounding schemes. In *ICML*, 2008.

[20] M. Schlesinger. Syntactic analysis of two-dimensional visual signals in noisy conditions. *Kibernetika*, 1976.

[21] R. Szeliski, R. Zabih, D. Scharstein, O. Veksler, V. Kolmogorov, A. Agarwala, M. Tappen, and C. Rother. A comparative study of energy minimization methods for Markov random fields with smoothness-based priors. *PAMI*, 2008.

| Image 1 | Image 2 | Ground Truth |

| BP | TRW | SWAP |
| Time=178.6s, Energy=4595612 | Time=512.0s, Energy=1851648 | Time=48.5s, Energy=1914655 |

| EXP | INT | HIER |
| Time=41.9s, Energy=1911774 | Time=2108.6s, Energy=1890418 | Time=363.2s, Energy=1873082 |

Figure 7: Results for the 'teddy' image pair with truncated quadratic pairwise potentials.

[22] D. Tarlow, D. Batra, P. Kohli, and V. Kolmogorov. Dynamic tree block coordinate ascent. In *ICML*, 2011.

[23] O. Veksler. *Efficient graph-based energy minimization methods in computer vision*. PhD thesis, Cornell University, 1999.

[24] O. Veksler. Graph cut based optimization for MRFs with truncated convex priors. In *CVPR*, 2007.

[25] M. Wainwright, T. Jaakkola, and A. Willsky. MAP estimation via agreement on trees: Message passing and linear programming. *Transactions on Information Theory*, 2005.

[26] Y. Weiss, C. Yanover, and T. Meltzer. MAP estimation, linear programming and belief propagation with convex free energies. In *UAI*, 2007.

[27] T. Werner. A linear programming approach to max-sum problem: A review. *PAMI*, 2007.

[28] T. Werner. Revisting the linear programming relaxation approach to Gibbs energy minimization and weighted constraint satisfaction. *PAMI*, 2010.

## Appendix A: Proof of Theorem 1

We first establish the theoretical property of the complete move-making algorithm using the following lemma.

**Lemma 1.** *The tight multiplicative bound of the complete move-making algorithm is equal to the submodular distortion of the distance function.*

*Proof.* The submodular distortion of a distance function $d$ is obtained by computing its tightest submodular overestimation as follows:

$$\overline{d} = \operatorname*{argmin}_{d'} t \tag{7}$$

$$\text{s.t.} \quad d'(l_i, l_j) \leq td(l_i, l_j), \forall l_i, l_j \in \mathbf{L},$$
$$d'(l_i, l_j) \geq d(l_i, l_j), \forall l_i, l_j \in \mathbf{L},$$
$$d'(l_i, l_j) + d'(l_{i+1}, l_{j+1}) \leq d'(l_i, l_{j+1}) + d'(l_{i+1}, l_j), \forall l_i, l_j \in \mathbf{L} \backslash \{l_h\}.$$

In order to prove the theorem, it is important to note that the definition of submodular distance function implies the following:

$$\overline{d}(l_i, l_j) + \overline{d}(l_{i'}, l_{j'}) \leq \overline{d}(l_i, l_{j'}) + \overline{d}(l_{i'}, l_j), \forall i' > i, j' > j.$$

A simple proof for the above claim can be found in [7].

We denote the submodular distortion of $d$ by $B$. By definition, it follows that

$$d(l_i, l_j) \leq \overline{d}(l_i, l_j) \leq Bd(l_i, l_j), \forall l_i, l_j \in \mathbf{L}. \tag{8}$$

We denote an optimal labeling of the original metric labeling problem as $\mathbf{x}^*$, that is,

$$\mathbf{x}^* = \operatorname*{argmin}_{\mathbf{x} \in \mathbf{L}^n} \sum_{X_a \in \mathbf{X}} \theta_a(x_a) + \sum_{(X_a, X_b) \in \mathbf{E}} w_{ab} d(x_a, x_b). \tag{9}$$

As the metric labeling problem is NP-hard, an optimal labeling $\mathbf{x}^*$ cannot be computed efficiently using any known algorithm. In order to obtain an approximate solution $\hat{\mathbf{x}}$, the complete move-making algorithm replaces the original distance function $d$ by its submodular overestimation $\overline{d}$, that is,

$$\hat{\mathbf{x}} = \operatorname*{argmin}_{\mathbf{x} \in \mathbf{L}^n} \sum_{X_a \in \mathbf{X}} \theta_a(x_a) + \sum_{(X_a, X_b) \in \mathbf{E}} w_{ab} \overline{d}(x_a, x_b). \tag{10}$$

Since the pairwise potentials in the above problem are submodular, the approximate solution $\hat{\mathbf{x}}$ can be obtained by solving a single minimum $st$-cut problem using the method of [7]. Using inequality (8), it follows that

$$\sum_{X_a \in \mathbf{X}} \theta_a(\hat{x}_a) + \sum_{(X_a, X_b) \in \mathbf{E}} w_{ab} d(\hat{x}_a, \hat{x}_b)$$

$$\leq \sum_{X_a \in \mathbf{X}} \theta_a(\hat{x}_a) + \sum_{(X_a, X_b) \in \mathbf{E}} w_{ab} \overline{d}(\hat{x}_a, \hat{x}_b)$$

$$\leq \sum_{X_a \in \mathbf{X}} \theta_a(x_a^*) + \sum_{(X_a, X_b) \in \mathbf{E}} w_{ab} \overline{d}(x_a^*, x_b^*)$$

$$\leq \sum_{X_a \in \mathbf{X}} \theta_a(x_a^*) + B \sum_{(X_a, X_b) \in \mathbf{E}} w_{ab} d(x_a^*, x_b^*).$$

The above inequality proves that the multiplicative bound of the complete move-making algorithm is at most $B$. It order to prove that it is exactly equal to $B$, we need to construct an example for which the bound is tight. To this end, let $l_k$ and $l_{k'}$ be two labels in the set $\mathbf{L}$ such that $k < k'$ and

$$\frac{\overline{d}(l_k, l_{k'})}{d(l_k, l_{k'})} = B.$$

Since $B$ is the minimum possible value of the maximum ratio of the estimated distance $\overline{d}$ to the original distance $d$, such a pair of labels must exist (otherwise, the submodular distortion can be

reduced further). Let us assume that there exists an $l_j \in \mathbf{L}$ such that $j < k$. Other cases (where $j > k'$ or $k < j < k'$) can be handled similarly. Note that since $\overline{d}$ is submodular, it follows that

$$\overline{d}(l_k, l_j) + \overline{d}(l_j, l_{k'}) \geq \overline{d}(l_k, l_{k'}). \tag{11}$$

We define a metric labeling problem over two random variables $X_a$ and $X_b$ connected by an edge with weight $w_{ab} = 1$. The unary potentials are defined as follows:

$$\theta_a(i) = \begin{cases} 0, & \text{if } i = k, \\ \frac{\overline{d}(l_k, l_{k'}) + \overline{d}(l_k, l_j) - \overline{d}(l_j, l_{k'})}{2}, & \text{if } i = j, \\ \infty & \text{otherwise}, \end{cases}$$

$$\theta_b(i) = \begin{cases} 0, & \text{if } i = k', \\ \frac{\overline{d}(l_k, l_{k'}) - \overline{d}(l_k, l_j) + \overline{d}(l_j, l_{k'})}{2}, & \text{if } i = j, \\ \infty & \text{otherwise}. \end{cases}$$

For the above metric labeling problem, it can be verified that an optimal solution $\mathbf{x}^*$ of problem (9) is the following: $x_a^* = l_k$ and $x_b^* = l_{k'}$. Furthermore, using inequality (11), it can be shown that the following is an optimal solution of problem (10): $\hat{x}_a = l_j$ and $\hat{x}_b = l_j$. In other words, $\hat{\mathbf{x}}$ is a valid approximate labeling provided by the complete move-making algorithm. The labelings $\mathbf{x}^*$ and $\hat{\mathbf{x}}$ satisfy the following equality:

$$\sum_{X_a \in \mathbf{X}} \theta_a(\hat{x}_a) + \sum_{(X_a, X_b) \in \mathbf{E}} w_{ab} d(\hat{x}_a, \hat{x}_b) = \sum_{X_a \in \mathbf{X}} \theta_a(x_a^*) + B \sum_{(X_a, X_b) \in \mathbf{E}} w_{ab} d(x_a^*, x_b^*).$$

Therefore, the tight multiplicative bound of the complete move-making algorithm is exactly equal to the submodular distortion of the distance function $d$. $\square$

We now turn our attention to the complete rounding procedure for the LP relaxation. Before we can establish its tight approximation factor, we need to compute the expected distance between the labels assigned to a pair of neighboring random variables. Recall that, in our notation, we denote a feasible solution of the LP relaxation by $\mathbf{y}$. For any feasible solution $\mathbf{y}$, we define $\mathbf{y}_a$ as the vector whose elements are the unary variables of $\mathbf{y}$ for the random variable $X_a \in \mathbf{X}$, that is,

$$\mathbf{y}_a = [y_a(i), \forall l_i \in \mathbf{L}]. \tag{12}$$

Similarly, we define $\mathbf{y}_{ab}$ as the vector whose elements are the pairwise variables of $\mathbf{y}$ for the neighboring random variables $(X_a, X_b) \in \mathbf{E}$, that is,

$$\mathbf{y}_{ab} = [y_{ab}(i, j), \forall l_i, l_j \in \mathbf{L}]. \tag{13}$$

Furthermore, using $\mathbf{y}_a$, we define $\mathbf{Y}_a$ as

$$Y_a(i) = \sum_{j=1}^{i} y_a(j).$$

In other words, if $\mathbf{y}_a$ is interpreted as the probability distribution over the labels of $X_a$, then $\mathbf{Y}_a$ is the corresponding cumulative distribution.

Given a feasible solution $\mathbf{y}$, we denote the integer solution obtained using the complete rounding procedure as $\hat{\mathbf{y}}$. The distance between the two labels encoded by vectors $\hat{\mathbf{y}}_a$ and $\hat{\mathbf{y}}_b$ will be denoted by $\hat{d}(\hat{\mathbf{y}}_a, \hat{\mathbf{y}}_b)$. In other words, if $\hat{f}_a$ and $\hat{f}_b$ are the indices of the labels assigned to $X_a$ and $X_b$ (that is, $\hat{y}_a(\hat{f}_a) = 1$ and $\hat{y}_b(\hat{f}_b) = 1$), then $\hat{d}(\hat{\mathbf{y}}_a, \hat{\mathbf{y}}_b) = d(l_{\hat{f}_a}, l_{\hat{f}_b})$.

The following shorthand notation would be useful for our analysis.

$$D_1(i) = \frac{1}{2} \left( d(l_i, l_1) + d(l_i, l_h) - d(l_{i+1}, l_1) - d(l_{i+1}, l_h) \right), \forall i \in \{1, \cdots, h-1\}, \tag{14}$$

$$D_2(i, j) = \frac{1}{2} \left( d(l_i, l_{j+1}) + d(l_{i+1}, l_j) - d(l_i, l_j) - d(l_{i+1}, l_{j+1}) \right), \forall i, j \in \{1, \cdots, h-1\}.$$

Using the above notation, we can state the following lemma on the expected distance of the rounded solution for two neighboring random variables.

**Lemma 2.** *Let* $\mathbf{y}$ *be a feasible solution of the* LP *relaxation,* $\mathbf{Y}_a$ *and* $\mathbf{Y}_b$ *be cumulative distributions of* $\mathbf{y}_a$ *and* $\mathbf{y}_b$, *and* $\hat{\mathbf{y}}$ *be the integer solution obtained by the complete rounding procedure for* $\mathbf{y}$. *Then, the following equation holds true:*

$$\mathbb{E}(\hat{d}(\hat{\mathbf{y}}_a, \hat{\mathbf{y}}_b)) = \sum_{i=1}^{h-1} Y_a(i) D_1(i) + \sum_{j=1}^{h-1} Y_b(j) D_1(j) + \sum_{i=1}^{h-1} \sum_{j=1}^{h-1} |Y_a(i) - Y_b(j)| D_2(i, j).$$

*Proof.* We define $\hat{f}_a$ and $\hat{f}_b$ to be the indices of the labels assigned to $X_a$ and $X_b$ by the rounded integer solution $\hat{\mathbf{y}}$. In other words, $\hat{y}_a(i) = 1$ if and only if $i = \hat{f}_a$ and $\hat{y}_b(j) = 1$ if and only if $j = \hat{f}_b$. We define binary variables $z_a(i)$ and $z_b(j)$ as follows:

$$z_a(i) = \left\{ \begin{array}{ll} 1 & \text{if } i \leq \hat{f}_a, \\ 0 & \text{otherwise,} \end{array} \right. \quad z_b(j) = \left\{ \begin{array}{ll} 1 & \text{if } j \leq \hat{f}_b, \\ 0 & \text{otherwise.} \end{array} \right.$$

For complete rounding, it can be verified that

$$\mathbb{E}(z_a(i)) = Y_a(i). \tag{15}$$

Furthermore, we also define binary variables $z_{ab}(i, j)$ that indicate whether $i$ and $j$ are contained within the interval defined by $\hat{f}_a$ and $\hat{f}_b$. Formally,

$$z_{ab}(i, j) = \left\{ \begin{array}{ll} 1 & \text{if } \min\{i, j\} \geq \min\{\hat{f}_a, \hat{f}_b\} \text{ and } \max\{i, j\} < \max\{\hat{f}_a, \hat{f}_b\}, \\ 0 & \text{otherwise.} \end{array} \right.$$

For complete rounding, it can be verified that

$$\mathbb{E}(z_{ab}(i, j)) = |Y_a(i) - Y_b(j)|. \tag{16}$$

Using the result of [7], we know that

$$\hat{d}(\hat{\mathbf{y}}_a, \hat{\mathbf{y}}_b) = \sum_{i=1}^{h-1} z_a(i) D_1(i) + \sum_{j=1}^{h-1} z_b(j) D_1(j) + \sum_{i=1}^{h-1} \sum_{j=1}^{h-1} z_{ab}(i, j) D_2(i, j).$$

The proof of the lemma follows by taking the expectation of the LHS and the RHS of the above equation and simplifying the RHS using the linearity of expectation and equations (15) and (16). $\square$

In order to state the next lemma, we require the definition of uncrossing pairwise variables. Given unary variables $\mathbf{y}_a$ and $\mathbf{y}_b$, the pairwise variable vector $\mathbf{y}'_{ab}$ is called uncrossing with respect to $\mathbf{y}_a$ and $\mathbf{y}_b$ if it satisfies the following properties:

$$\sum_{j=1}^{h} y'_{ab}(i, j) = y_a(i), \forall i \in \{1, 2, \cdots, h\},$$

$$\sum_{i=1}^{h} y'_{ab}(i, j) = y_b(i), \forall j \in \{1, 2, \cdots, h\},$$

$$y'_{ab}(i, j) \geq 0, \forall i, j \in \{1, 2, \cdots, h\},$$

$$\min\{y'_{ab}(i, j'), y'_{ab}(i', j)\} = 0, \forall i, j, i', j' \in \{1, 2, \cdots, h\}, i < i', j < j'. \tag{17}$$

The following lemma establishes a connection between the expected distance between the labels assigned by complete rounding and the pairwise cost specified by uncrossing pairwise variables.

**Lemma 3.** *Let* $\mathbf{y}$ *be a feasible solution of the* LP *relaxation, and* $\hat{\mathbf{y}}$ *be the integer solution obtained by the complete rounding procedure for* $\mathbf{y}$. *Furthermore, let* $\mathbf{y}'_{ab}$ *be uncrossing pairwise variables with respect to* $\mathbf{y}_a$ *and* $\mathbf{y}_b$. *Then, the following equation holds true:*

$$\mathbb{E}(\hat{d}(\hat{\mathbf{y}}_a, \hat{\mathbf{y}}_b)) = \sum_{i=1}^{h} \sum_{j=1}^{h} d(l_i, l_j) y'_{ab}(i, j).$$

*Proof.* We define $\mathbf{Y}_a$ and $\mathbf{Y}_b$ to be the cumulative distributions corresponding to $\mathbf{y}_a$ and $\mathbf{y}_b$ respectively. We claim that the uncrossing property (17) implies the following condition:

$$\sum_{i=1}^{i'}\sum_{j=1}^{j'} y'_{ab}(i,j) = \min\{Y_a(i'), Y_b(j')\}, \forall i', j' \in \{1, \cdots, h\}. \tag{18}$$

To prove this claim, assume that $Y_a(i') < Y_b(j')$. The other cases can be handled similarly. Since $\mathbf{y}'_{ab}$ satisfies the constraints of the LP relaxation, it follows that:

$$\sum_{i=1}^{h}\sum_{j=1}^{j'} y'_{ab}(i,j) = Y_b(j'),$$

$$\sum_{i=1}^{i'}\sum_{j=1}^{h} y'_{ab}(i,j) = Y_a(i'). \tag{19}$$

Since the LHS of equality (18) is less than or equal to the LHS of both the above equations, it follows that

$$\sum_{i=1}^{i'}\sum_{j=1}^{j'} y'_{ab}(i,j) \le \min\{Y_a(i'), Y_b(j')\}. \tag{20}$$

Therefore, there must exist a $k > i'$ and $k' \le j'$ such that $y'_{ab}(k, k') \neq 0$. Otherwise, the LHS in the above inequality will be exactly equal to $Y_b(j')$, which would result in a contradiction. By the uncrossing property (17), we know that $\min\{y'_{ab}(i,j), y'_{ab}(k,k')\} = 0$ if $i \le i'$ and $j > j'$. Therefore, $y'_{ab}(i,j) = 0$ for all $i \le i'$ and $j > j'$, which proves the claim.

Combining equations (19) and (20), we get the following:

$$\sum_{i=1}^{i'}\sum_{j=j'+1}^{h} y'_{ab}(i,j) + \sum_{i=i'+1}^{h}\sum_{j=1}^{j'} y'_{ab}(i,j) = |Y_a(i) - Y_b(j)|, \forall i', j' \in \{1, \cdots, h\}.$$

By solving for $\mathbf{y}'_{ab}$ using the above equations, we get

$$\sum_{i=1}^{h}\sum_{j=1}^{h} d(l_i, l_j) y'_{ab}(i,j) = \sum_{i=1}^{h-1} Y_a(i) D_1(i) + \sum_{j=1}^{h-1} Y_b(j) D_1(j) + \sum_{i=1}^{h-1}\sum_{j=1}^{h-1} |Y_a(i) - Y_b(j)| D_2(i,j).$$

Using the previous lemma, this proves that

$$\mathbb{E}(\hat{d}(\hat{\mathbf{y}}_a, \hat{\mathbf{y}}_b)) = \sum_{i=1}^{h}\sum_{j=1}^{h} d(l_i, l_j) y'_{ab}(i,j).$$

$\square$

Our next lemma establishes that uncrossing pairwise variables are optimal for submodular distance functions.

**Lemma 4.** *Let $\mathbf{y}'_{ab}$ be the uncrossing pairwise variables with respect to the unary variables $\mathbf{y}_a$ and $\mathbf{y}_b$. Let $\overline{d}: \mathbf{L} \times \mathbf{L} \to \mathbb{R}^+$ be a submodular distance function. Then the following condition holds true:*

$$\mathbf{y}'_{ab} = \underset{\overline{\mathbf{y}}_{ab}}{\operatorname{argmin}} \sum_{i=1}^{h}\sum_{j=1}^{h} \overline{d}(i,j) \overline{y}_{ab}(i,j), \tag{21}$$

$$s.t. \quad \sum_{j=1}^{h} \overline{y}_{ab}(i,j) = y_a(i), \forall i \in \{1, \cdots, h\},$$

$$\sum_{i=1}^{h} \overline{y}_{ab}(i,j) = y_b(j), \forall j \in \{1, \cdots, h\},$$

$$\overline{y}_{ab}(i,j) \ge 0, \forall i, j \in \{1, \cdots, h\}.$$

*Proof.* We prove the lemma by contradiction. Suppose that the optimal solution to the above problem is $\mathbf{y}''_{ab}$, which is not uncrossing. Let

$$\min\{y''_{ab}(i,j'), y''_{ab}(i',j)\} = \lambda \neq 0,$$

where $i < i'$ and $j < j'$. Since $\overline{d}$ is submodular, it implies that

$$\overline{d}(l_i, l_j) + \overline{d}(l_{i'}, l_{j'}) \leq \overline{d}(l_i, l_{j'}) + \overline{d}(l_{i'}, l_j).$$

Therefore the objective function of problem (21) can be reduced further by the following modification:

$$y''_{ab}(i,j') \leftarrow y''_{ab}(i,j') - \lambda, y''_{ab}(i',j) \leftarrow y''_{ab}(i',j) - \lambda,$$
$$y''_{ab}(i,j) \leftarrow y''_{ab}(i,j) + \lambda, y''_{ab}(i',j') \leftarrow y''_{ab}(i',j') + \lambda.$$

The resulting contradiction proves our claim that the uncrossing pairwise variables $\mathbf{y}'_{ab}$ are an optimal solution of problem (21). $\qquad\square$

Using the above lemmas, we will now obtain the tight approximation factor of the complete rounding procedure.

**Lemma 5.** *The tight approximation factor of the complete rounding procedure is equal to the submodular distortion of the distance function.*

*Proof.* We denote a feasible fractional solution of the LP relaxation by $\mathbf{y}$ and the rounded solution by $\hat{\mathbf{y}}$. Consider a pair of neighboring random variables $(X_a, X_b) \in \mathbf{X}$. We define uncrossing pairwise variables $\mathbf{y}'_{ab}$ with respect to $\mathbf{y}_a$ and $\mathbf{y}_b$. Using lemmas 3 and 4, the approximation factor of the complete rounding procedure can be shown to be at most $B$ as follows:

$$
\begin{aligned}
\mathbb{E}(\hat{d}(\hat{\mathbf{y}}_a, \hat{\mathbf{y}}_b)) &= \sum_{i=1}^{h}\sum_{j=1}^{h} d(l_i, l_j) y'_{ab}(i,j) \\
&\leq \sum_{i=1}^{h}\sum_{j=1}^{h} \overline{d}(l_i, l_j) y'_{ab}(i,j) \\
&\leq \sum_{i=1}^{h}\sum_{j=1}^{h} \overline{d}(l_i, l_j) y_{ab}(i,j) \\
&\leq B\sum_{i=1}^{h}\sum_{j=1}^{h} d(l_i, l_j) y_{ab}(i,j).
\end{aligned}
$$

In order to prove that the approximation factor of the complete rounding is exactly $B$, we need an example where the above inequality holds as an equality. The key to obtaining a tight example lies in the Lagrangian dual of problem (7). In order to specify its dual, we need three types of dual variables. The first type, denoted by $\alpha(i,j)$, corresponds to the constraint

$$d'(l_i, l_j) \leq td(l_i, l_j).$$

The second type, denoted by $\beta(i,j)$, corresponds to the constraint

$$d'(l_i, l_j) \geq d(l_i, l_j).$$

The third type, denoted by $\gamma(i,j)$, corresponds to the constraint

$$d'(l_i, l_j) + d'(l_{i+1}, l_{j+1}) \leq d'(l_i, l_{j+1}) + d'(l_{i+1}, l_j).$$

Using the above variables, the dual of problem (7) is given by

$$\max \quad \sum_{i=1}^{h}\sum_{j=1}^{h} d(l_i, l_j)\beta(i,j) \qquad (22)$$

$$
\begin{aligned}
\text{s.t.} \quad & \sum_{i=1}^{h}\sum_{j=1}^{h} d(l_i, l_j)\alpha(i,j) = 1, \\
& \beta(i,j) = \alpha(i,j) - \gamma(i,j-1) - \gamma(i-1,j) + \gamma(i-1,j-1) + \gamma(i,j), \\
& \forall i,j \in \{1,\cdots,h\}, \\
& \alpha(i,j) \geq 0, \beta(i,j) \geq 0, \gamma(i,j) \geq 0, \forall i,j \in \{1,\cdots,h\}.
\end{aligned}
$$

We claim that the above dual problem has an optimal solution $(\boldsymbol{\alpha}^*, \boldsymbol{\beta}^*, \boldsymbol{\gamma}^*)$ that satisfies the following property:

$$\min\{\beta^*(i, j'), \beta^*(i', j)\} = 0, \forall i, i', j, j' \in \{1, \cdots, h\}, i < i', j < j'. \tag{23}$$

We refer to the optimal dual solution $\boldsymbol{\beta}^*$ that satisfies the above property as uncrossing dual variables as it is analogous to uncrossing pairwise variables. The above claim, namely, the existence of an uncrossing optimal dual solution, can be proved by contradiction as follows. Suppose there exists no optimal solution that satisfies the above property. Then consider the following problem, which is the dual of the problem of finding the tightest submodular overestimate of the submodular function $\bar{d}$:

$$\max \quad \sum_{i=1}^{h} \sum_{j=1}^{h} \bar{d}(l_i, l_j) \beta(i, j) \tag{24}$$

$$\text{s.t.} \quad \sum_{i=1}^{h} \sum_{j=1}^{h} \bar{d}(l_i, l_j) \alpha(i, j) = 1,$$

$$\beta(i, j) = \alpha(i, j) - \gamma(i, j - 1) - \gamma(i - 1, j) + \gamma(i - 1, j - 1) + \gamma(i, j),$$

$$\forall i, j \in \{1, \cdots, h\},$$

$$\alpha(i, j) \geq 0, \beta(i, j) \geq 0, \gamma(i, j) \geq 0, \forall i, j \in \{1, \cdots, h\}.$$

By strong duality, problem (24) has an optimal value of 1. However, the optimal solution of problem (22), which is also a feasible solution for problem (24), provides a value strictly greater than 1. This results in a contradiction that proves our claim.

The optimal dual variables that satisfy property (23) allow us to construct an example that proves that the approximation factor $B$ of the complete rounding procedure is tight. Specifically, we define

$$\bar{y}_{ab}(i, j) = \frac{\alpha^*(i, j)}{\sum_{i'=1}^{h} \sum_{j'=1}^{h} \alpha^*(i', j')}, \forall i, j \in \{1, \cdots, h\},$$

$$\bar{y}_a(i) = \sum_{j=1}^{h} \bar{y}_{ab}(i, j), \forall i \in \{1, \cdots, h\},$$

$$\bar{y}_b(j) = \sum_{i=1}^{h} \bar{y}_{ab}(i, j), \forall j \in \{1, \cdots, h\}.$$

Note that the pairwise variables $\bar{\mathbf{y}}_{ab}$ must minimize the pairwise potential corresponding to the unary variables $\mathbf{y}_a$ and $\mathbf{y}_b$, that is,

$$\bar{\mathbf{y}}_{ab} = \underset{\mathbf{y}_{ab}}{\operatorname{argmin}} \sum_{l_i, l_j \in \mathbf{L}} d(l_i, l_j) y_{ab}(i, j) \tag{25}$$

$$\text{s.t.} \quad \sum_{l_j \in \mathbf{L}} y_{ab}(i, j) = \bar{y}_a(i), \forall l_i \in \mathbf{L}$$

$$\sum_{l_i \in \mathbf{L}} y_{ab}(i, j) = \bar{y}_b(j), \forall l_j \in \mathbf{L}$$

$$y_{ab}(i, j) \geq 0, \forall l_i, l_j \in \mathbf{L}.$$

If the above statement was not true, then the value of the dual problem (22) could be increased further.

We also define the following pairwise variables:

$$y'_{ab}(i, j) = \frac{\beta^*(i, j)}{\sum_{i'=1}^{h} \sum_{j'=1}^{h} \beta^*(i', j')}, \forall i, j \in \{1, \cdots, h\},$$

$$y'_a(i) = \sum_{j=1}^{h} y'_{ab}(i, j), \forall i \in \{1, \cdots, h\},$$

$$y'_b(j) = \sum_{i=1}^{h} y'_{ab}(i, j), \forall j \in \{1, \cdots, h\}.$$

It can be verified that

$$y'_a(i) = \overline{y}_a(i), y'_b(j) = \overline{y}_b(j), \forall i, j \in \{1, \cdots, h\}.$$

The above condition follows from the constraints of problem (22). Due to the uncrossing property of $\boldsymbol{\beta}^*$, the pairwise variables $\mathbf{y}'_{ab}$ are uncrossing with respect to $\overline{y}_a$ and $\overline{y}_b$. By lemma 3, this implies that

$$\mathbb{E}(\hat{d}(\hat{\mathbf{y}}_a, \hat{\mathbf{y}}_b)) = \sum_{i=1}^{h} \sum_{j=1}^{h} d(l_i, l_j) y'_{ab}(i, j),$$

where $\hat{\mathbf{y}}_a$ and $\hat{\mathbf{y}}_b$ are integer solutions obtained by the complete rounding procedure. By strong duality, it follows that

$$\mathbb{E}(\hat{d}(\hat{\mathbf{y}}_a, \hat{\mathbf{y}}_b)) = B \sum_{i=1}^{h} \sum_{j=1}^{h} d(l_i, l_j) \overline{y}_{ab}(i, j). \tag{26}$$

The existence of an example that satisfies properties (25) and (26) implies that the tight approximation factor of the complete rounding procedure is $B$. □

Lemmas 1 and 5 together prove theorem 1.

## Appendix B: Proof of Theorem 2

We begin by establishing the theoretical properties of the interval-move making algorithm. Recall that, given an interval of labels $\mathbf{I} = \{l_s, \cdots, l_e\}$ of at most $q$ consecutive labels and a labeling $\hat{x}$, we define $\mathbf{I}_a = \mathbf{I} \cup \{\hat{x}_a\}$ for all random variables $X_a \in \mathbf{X}$. In order to use the interval move-making algorithm, we compute a submodular distance function $\overline{d}_{\hat{x}_a, \hat{x}_b} : \mathbf{I}_a \times \mathbf{I}_b \to \mathbb{R}^+$ for all pairs of neighboring random variables $(X_a, X_b) \in \mathbf{E}$ as follows:

$$\overline{d}_{\hat{x}_a, \hat{x}_b} = \underset{d'}{\operatorname{argmin}} \, t \tag{27}$$

$$\text{s.t.} \quad d'(l_i, l_j) \le td(l_i, l_j), \forall l_i \in \mathbf{I}_a, l_j \in \mathbf{I}_b,$$
$$d'(l_i, l_j) \ge d(l_i, l_j), \forall l_i \in \mathbf{I}_a, l_j \in \mathbf{I}_b,$$
$$d'(l_i, l_j) + d'(l_{i+1}, l_{j+1}) \le d'(l_i, l_{j+1}) + d'(l_{i+1}, l_j), \forall l_i, l_j \in \mathbf{I}\backslash\{l_e\},$$
$$d'(l_i, l_e) + d'(l_{i+1}, \hat{x}_b) \le d'(l_i, \hat{x}_b) + d'(l_{i+1}, l_e), \forall l_i \in \mathbf{I}\backslash\{l_e\},$$
$$d'(l_e, l_j) + d'(\hat{x}_a, l_{j+1}) \le d'(l_e, l_{j+1}) + d'(\hat{x}_a, l_j), \forall l_j \in \mathbf{I}\backslash\{l_e\},$$
$$d'(l_e, l_e) + d(\hat{x}_a, \hat{x}_b) \le d'(l_e, \hat{x}_b) + d'(\hat{x}_a, l_e).$$

For any interval $\mathbf{I}$ and labeling $\mathbf{x}$, we define the following sets:

$$\mathbf{V}(\mathbf{x}, \mathbf{I}) = \{X_a | X_a \in \mathbf{X}, x_a \in \mathbf{I}\},$$
$$\mathbf{A}(\mathbf{x}, \mathbf{I}) = \{(X_a, X_b) | (X_a, X_b) \in \mathbf{E}, x_a \in \mathbf{I}, x_b \in \mathbf{I}\},$$
$$\mathbf{B}_1(\mathbf{x}, \mathbf{I}) = \{(X_a, X_b) | (X_a, X_b) \in \mathbf{E}, x_a \in \mathbf{I}, x_b \notin \mathbf{I}\},$$
$$\mathbf{B}_2(\mathbf{x}, \mathbf{I}) = \{(X_a, X_b) | (X_a, X_b) \in \mathbf{E}, x_a \notin \mathbf{I}, x_b \in \mathbf{I}\},$$
$$\mathbf{B}(\mathbf{x}, \mathbf{I}) = \mathbf{B}_1(\mathbf{x}) \cup \mathbf{B}_2(\mathbf{x}).$$

In other words, $\mathbf{V}(\mathbf{x}, \mathbf{I})$ is the set of all random variables whose label belongs to the interval $\mathbf{I}$. Similarly, $\mathbf{A}(\mathbf{x}, \mathbf{I})$ is the set of all neighboring random variables such that the labels assigned to both the random variables belong to the interval $\mathbf{I}$. The set $\mathbf{B}(\mathbf{x}, \mathbf{I})$ contains the set of all neighboring random variables such that only one of the two labels assigned to the two random variables belongs to the interval $\mathbf{I}$. Given the set of all intervals $\mathcal{I}$ and a labeling $\hat{\mathbf{x}}$, we define the following for all $x_a, x_b \in \mathbf{L}$:

$$D(x_a, x_b; \hat{x}_a, \hat{x}_b) = \sum_{\mathbf{I} \in \mathcal{I}, \mathbf{A}(\mathbf{x}, \mathbf{I}) \ni (X_a, X_b)} d_{\hat{x}_a, \hat{x}_b}(x_a, x_b)$$
$$+ \sum_{\mathbf{I} \in \mathcal{I}, \mathbf{B}_1(\mathbf{x}, \mathbf{I}) \ni (X_a, X_b)} d_{\hat{x}_a, \hat{x}_b}(x_a, \hat{x}_b)$$
$$+ \sum_{\mathbf{I} \in \mathcal{I}, \mathbf{B}_2(\mathbf{x}, \mathbf{I}) \ni (X_a, X_b)} d_{\hat{x}_a, \hat{x}_b}(\hat{x}_a, x_b).$$

Using the above notation, we are ready to state the following lemma on the theoretical guarantee of the interval move-making algorithm.

**Lemma 6.** *The tight multiplicative bound of the interval move-making algorithm is equal to*

$$\frac{1}{q} \max_{x_a, x_b, \hat{x}_a, \hat{x}_b \in \mathbf{L}, x_a \neq x_b} \frac{D(x_a, x_b; \hat{x}_a, \hat{x}_b)}{d(x_a, x_b)}.$$

*Proof.* We denote an optimal labeling by $\mathbf{x}^*$ and the estimated labeling by $\hat{\mathbf{x}}$. Let $t \in [1, q]$ be a uniformly distributed random integer. Using $t$, we define the following set of non-overlapping intervals:

$$\mathcal{I}_t = \{[1, t], [t+1, t+q], \cdots, [., h]\}.$$

For each interval $\mathbf{I} \in \mathcal{I}_t$, we define a labeling $\mathbf{x}^{\mathbf{I}}$ as follows:

$$x_a^{\mathbf{I}} = \begin{cases} x_a^* & \text{if } x_a^* \in \mathbf{I}, \\ \hat{x}_a & \text{otherwise.} \end{cases}$$

Since $\hat{\mathbf{x}}$ is the labeling obtained after the interval move-making algorithm converges, it follows that

$$\sum_{X_a \in \mathbf{X}} \theta_a(\hat{x}_a) + \sum_{(X_a, X_b) \in \mathbf{E}} w_{ab} \overline{d}_{\hat{x}_a, \hat{x}_b}(\hat{x}_a, \hat{x}_b) \leq \sum_{X_a \in \mathbf{X}} \theta_a(x_a^{\mathbf{I}}) + \sum_{(X_a, X_b) \in \mathbf{E}} w_{ab} \overline{d}_{\hat{x}_a, \hat{x}_b}(x_a^{\mathbf{I}}, x_b^{\mathbf{I}}).$$

By canceling out the common terms, we can simplify the above inequality as

$$\sum_{X_a \in \mathbf{V}(\mathbf{x}^*, \mathbf{I})} \theta_a(\hat{x}_a)$$

$$+ \sum_{(X_a, X_b) \in \mathbf{A}(\mathbf{x}^*, \mathbf{I})} w_{ab} \overline{d}_{\hat{x}_a, \hat{x}_b}(\hat{x}_a, \hat{x}_b)$$

$$+ \sum_{(X_a, X_b) \in \mathbf{B}_1(\mathbf{x}^*, \mathbf{I})} w_{ab} \overline{d}_{\hat{x}_a, \hat{x}_b}(\hat{x}_a, \hat{x}_b)$$

$$+ \sum_{(X_a, X_b) \in \mathbf{B}_2(\mathbf{x}^*, \mathbf{I})} w_{ab} \overline{d}_{\hat{x}_a, \hat{x}_b}(\hat{x}_a, \hat{x}_b)$$

$$\leq \sum_{X_a \in \mathbf{V}(\mathbf{x}^*, \mathbf{I})} \theta_a(x_a^*)$$

$$+ \sum_{(X_a, X_b) \in \mathbf{A}(\mathbf{x}^*, \mathbf{I})} w_{ab} \overline{d}_{\hat{x}_a, \hat{x}_b}(x_a^*, x_b^*)$$

$$+ \sum_{(X_a, X_b) \in \mathbf{B}_1(\mathbf{x}^*, \mathbf{I})} w_{ab} \overline{d}_{\hat{x}_a, \hat{x}_b}(x_a^*, \hat{x}_b)$$

$$+ \sum_{(X_a, X_b) \in \mathbf{B}_2(\mathbf{x}^*, \mathbf{I})} w_{ab} \overline{d}_{\hat{x}_a, \hat{x}_b}(\hat{x}_a, x_b^*).$$

We now sum the above inequality over all the intervals $\mathbf{I} \in \mathcal{I}_t$. Note that the resulting LHS is at least equal to the energy of the labeling $\hat{\mathbf{x}}$ when the distance function between the random variables $(X_a, X_b)$ is $\overline{d}_{\hat{x}_a, \hat{x}_b}$. This implies that

$$\sum_{X_a \in \mathbf{X}} \theta_a(\hat{x}_a) + \sum_{(X_a, X_b) \in \mathbf{E}} w_{ab} \overline{d}_{\hat{x}_a, \hat{x}_b}(\hat{x}_a, \hat{x}_b)$$

$$\leq \sum_{X_a \in \mathbf{X}} \theta_a(x_a^*)$$

$$+ \sum_{\mathbf{I} \in \mathcal{I}_t} \sum_{(X_a, X_b) \in \mathbf{A}(\mathbf{x}^*, \mathbf{I})} w_{ab} \overline{d}_{\hat{x}_a, \hat{x}_b}(x_a^*, x_b^*)$$

$$+ \sum_{\mathbf{I} \in \mathcal{I}_t} \sum_{(X_a, X_b) \in \mathbf{B}_1(\mathbf{x}^*, \mathbf{I})} w_{ab} \overline{d}_{\hat{x}_a, \hat{x}_b}(x_a^*, \hat{x}_b)$$

$$+ \sum_{\mathbf{I} \in \mathcal{I}_t} \sum_{(X_a, X_b) \in \mathbf{B}_2(\mathbf{x}^*, \mathbf{I})} w_{ab} \overline{d}_{\hat{x}_a, \hat{x}_b}(\hat{x}_a, x_b^*).$$

Taking the expectation on both sides of the above inequality with respect to the uniformly distributed random integer $t \in [1, q]$ proves that the multiplicative bound of the interval move-making algorithm is at most equal to

$$\frac{1}{q} \max_{x_a, x_b, \hat{x}_a, \hat{x}_b \in \mathbf{L}, x_a \neq x_b} \frac{D(x_a, x_b; \hat{x}_a, \hat{x}_b)}{d(x_a, x_b)}.$$

A tight example with two random variables $X_a$ and $X_b$ with $w_{ab} = 1$ can be constructed similar to the one shown in lemma 1. $\qquad\square$

We now turn our attention to the interval rounding procedure. Let $\mathbf{y}$ be a feasible solution of the LP relaxation, and $\hat{\mathbf{y}}$ be the integer solution obtained using interval rounding. Once again, we define the unary variable vector $\mathbf{y}_a$ and the pairwise variable vector $\mathbf{y}_{ab}$ as specified in equations (12) and (13) respectively. Similar to the previous appendix, we denote the expected distance between $\hat{\mathbf{y}}_a$ and $\hat{\mathbf{y}}_b$ as $\hat{d}(\hat{\mathbf{y}}_a, \hat{\mathbf{y}}_b)$.

Given an interval $\mathbf{I} = \{l_s, \cdots, l_e\}$ of at most $q$ consecutive labels, we define a vector $\mathbf{Y}_a^{\mathbf{I}}$ for each random variable $X_a$ as follows:

$$Y_a^{\mathbf{I}}(i) = \sum_{j=s}^{s+i-1} y_a(j), \forall i \in \{1, \cdots, e-s+1\}.$$

In other words, $\mathbf{Y}_a^{\mathbf{I}}$ is the cumulative distribution of $\mathbf{y}_a$ within the interval $\mathbf{I}$. Furthermore, for each pair of neighboring random variables $(X_a, X_b) \in \mathbf{E}$ we define

$$Z_{ab}^{\mathbf{I}} = \max\{Y_a^{\mathbf{I}}(e-s+1), Y_b^{\mathbf{I}}(e-s+1)\} - \min\{Y_a^{\mathbf{I}}(e-s+1), Y_b^{\mathbf{I}}(e-s+1)\},$$
$$Z_a^{\mathbf{I}}(i) = \min\{Y_a^{\mathbf{I}}(i), Y_b^{\mathbf{I}}(e-s+1)\}, \forall i \in \{1, \cdots, e-s+1\},$$
$$Z_b^{\mathbf{I}}(j) = \min\{Y_b^{\mathbf{I}}(j), Y_a^{\mathbf{I}}(e-s+1)\}, \forall j \in \{1, \cdots, e-s+1\}.$$

The following shorthand notation would be useful to concisely specify the exact form of $\hat{d}(\hat{\mathbf{y}}_a, \hat{\mathbf{y}}_b)$.

$$D_0^{\mathbf{I}} = \max_{l_i, l_j, |\{l_i, l_j\} \cap \mathbf{I}| = 1} d(l_i, l_j),$$

$$D_1^{\mathbf{I}}(i) = \frac{1}{2} \left( d(l_{s+i-1}, l_s) + d(l_{s+i-1}, l_e) - d(l_{s+i}, l_s) - d(l_{s+i}, l_e) \right), \forall i \in \{1, \cdots, e-s\},$$

$$D_2^{\mathbf{I}}(i, j) = \frac{1}{2} \left( d(l_{s+i-1}, l_{s+j}) + d(l_{s+i}, l_{s+j-1}) - d(l_{s+i-1}, l_{s+j-1}) - d(l_{s+i}, l_{s+j}) \right),$$
$$\forall i, j \in \{1, \cdots, e-s\}.$$

In other words, $\mathbf{D}_0^{\mathbf{I}}$ is the maximum distance between two labels such that only one of the two labels lies in the interval $\mathbf{I}$. The terms $\mathbf{D}_1^{\mathbf{I}}$ and $\mathbf{D}_2^{\mathbf{I}}$ are analogous to the terms defined in equation (14). Using the above notation, we can state the following lemma on the expected distance of the rounded solution for two neighboring random variables.

**Lemma 7.** *Let $\mathbf{y}$ be a feasible solution of the* LP *relaxation, and $\hat{\mathbf{y}}$ be the integer solution obtained by the interval rounding procedure for $\mathbf{y}$ using the set of intervals $\mathcal{I}$. Then, the following inequality holds true:*

$$\mathbb{E}(\hat{d}(\hat{\mathbf{y}}_a, \hat{\mathbf{y}}_b)) \leq \frac{1}{q} \left( \sum_{\mathbf{I} = \{l_s, \cdots, l_e\} \in \mathcal{I}} Z_{ab}^{\mathbf{I}} D_0^{\mathbf{I}} + \sum_{i=1}^{e-s} Z_a^{\mathbf{I}}(i) D_1^{\mathbf{I}}(i) + \sum_{j=1}^{e-s} Z_b^{\mathbf{I}}(j) D_1^{\mathbf{I}}(j) + \right.$$
$$\left. \sum_{i=1}^{e-s} \sum_{j=1}^{e-s} |Z_a^{\mathbf{I}}(i) - Z_b^{\mathbf{I}}(j)| D_2^{\mathbf{I}}(i, j) \right).$$

*Proof.* We begin the proof by establishing the probability of a random variable $X_a$ being assigned a label in an iteration of the interval rounding procedure. The total number of intervals in the set $\mathcal{I}$ is $h + q - 1$. Out of all the intervals, each label $l_i$ is present in $q$ intervals. Thus, the probability of choosing an interval that contains the label $l_i$ is $q/(h + q - 1)$. Once an interval containing $l_i$ is chosen, the probability of assigning the label $l_i$ to $X_a$ is $y_a(i)$. Thus, the probability of assigning a

label $l_i$ to $X_a$ is $y_a(i)q/(h+q-1)$. Summing over all labels $l_i$, we observe that the probability of assigning a label to $X_a$ in an iteration of the interval rounding procedure is $q/(h+q-1)$.

Now we consider two neighboring random variables $(X_a, X_b) \in \mathbf{E}$. In the current iteration of interval rounding, given an interval $\mathbf{I}$, the probability of exactly one of the two random variables getting assigned a label in $\mathbf{I}$ is exactly equal to $Z_{ab}^{\mathbf{I}}$. In this case, we have to assume that the expected distance between the two variables will be the maximum possible, that is, $D_0^{\mathbf{I}}$. If both the variables are assigned a label in $\mathbf{I}$, then a slight modification of lemma 2 gives us the expected distance as

$$\sum_{i=1}^{e-s} Z_a^{\mathbf{I}}(i)D_1^{\mathbf{I}}(i) + \sum_{j=1}^{e-s} Z_b^{\mathbf{I}}(j)D_1^{\mathbf{I}}(j) + \sum_{i=1}^{e-s}\sum_{j=1}^{e-s} |Z_a^{\mathbf{I}}(i) - Z_b^{\mathbf{I}}(j)|D_2^{\mathbf{I}}(i,j).$$

Thus, the expected distance between the labels of $X_a$ and $X_b$ conditioned on at least one of the two random variables getting assigned in an iteration is equal to

$$\frac{1}{(h+q-1)}\left( \sum_{\mathbf{I}=\{l_s,\cdots,l_e\}\in\mathcal{I}} Z_{ab}^{\mathbf{I}}D_0^{\mathbf{I}} + \sum_{i=1}^{e-s} Z_a^{\mathbf{I}}(i)D_1^{\mathbf{I}}(i) + \sum_{j=1}^{e-s} Z_b^{\mathbf{I}}(j)D_1^{\mathbf{I}}(j)+ \right.$$
$$\left. \sum_{i=1}^{e-s}\sum_{j=1}^{e-s} |Z_a^{\mathbf{I}}(i) - Z_b^{\mathbf{I}}(j)|D_2^{\mathbf{I}}(i,j) \right),$$

where the term $1/(h+q-1)$ corresponds to the probability of choosing an interval from the set $\mathcal{I}$. In order to compute $\hat{d}(\hat{\mathbf{y}}_a, \hat{\mathbf{y}}_b)$, we need to divide the above term with the probability of at least one of the two random variables getting assigned a label in the current iteration. Since the two cumulative distributions $\mathbf{Y}_a^{\mathbf{I}}$ and $\mathbf{Y}_b^{\mathbf{I}}$ can be arbitrarily close to each other without being exactly equal, we can only lower bound the probability of at least one of $X_a$ and $X_b$ being assigned a label at the current iteration by the probability of $X_a$ being assigned a label in the current iteration. Since the probability of $X_a$ being assigned a label is $q/(h+q-1)$, it follows that

$$\mathbb{E}(\hat{d}(\hat{\mathbf{y}}_a, \hat{\mathbf{y}}_b)) \leq \quad \frac{1}{q}\left( \sum_{\mathbf{I}=\{l_s,\cdots,l_e\}\in\mathcal{I}} Z_{ab}^{\mathbf{I}}D_0^{\mathbf{I}} + \sum_{i=1}^{e-s} Z_a^{\mathbf{I}}(i)D_1^{\mathbf{I}}(i) + \sum_{j=1}^{e-s} Z_b^{\mathbf{I}}(j)D_1^{\mathbf{I}}(j)+ \right.$$
$$\left. \sum_{i=1}^{e-s}\sum_{j=1}^{e-s} |Z_a^{\mathbf{I}}(i) - Z_b^{\mathbf{I}}(j)|D_2^{\mathbf{I}}(i,j) \right).$$

$\square$

Using the above lemma, we can now establish the theoretical property of the interval rounding procedure.

**Lemma 8.** *The tight approximation factor of the interval rounding procedure is equal to*

$$\frac{1}{q} \max_{x_a,x_b,\hat{x}_a,\hat{x}_b\in\mathbf{L}, x_a\neq x_b} \frac{D(x_a, x_b; \hat{x}_a, \hat{x}_b)}{d(x_a, x_b)}.$$

*Proof.* Similar to lemma 5, the key to proving the above lemma lies in the dual of problem (27). Given the current labels $\hat{x}_a$ and $\hat{x}_b$ of the random variables $X_a$ and $X_b$ respectively, as well as an interval $\mathbf{I} = \{l_s, \cdots, l_e\}$, problem (27) computes the tightest submodular overestimation $\overline{d}_{\hat{x}_a, \hat{x}_b}$ : $\mathbf{I}_a \times \mathbf{I}_b \to \mathbb{R}^+$ where $\mathbf{I}_a = \mathbf{I} \cup \{\hat{x}_a\}$ and $\mathbf{I}_b = \mathbf{I} \cup \{\hat{x}_b\}$. Similar to problem (22), the dual of problem (27) consists of three types of dual variables. The first type, denoted by $\alpha(i, j; \hat{x}_a, \hat{x}_b, \mathbf{I})$ corresponds to the constraint

$$d'(l_i, l_j) \leq td(l_i, l_j).$$

The second type, denoted by $\beta(i, j; \hat{x}_b, \hat{x}_b, \mathbf{I})$, corresponds to the constraint

$$d'(l_i, l_j) \geq d(l_i, l_j).$$

The third type, denoted by $\gamma(i,j;\hat{x}_a,\hat{x}_b,\mathbf{I})$, corresponds to the constraint
$$d'(l_i,l_j) + d'(l_{i+1},l_{j+1}) \leq d'(l_i,l_{j+1}) + d'(l_{i+1},l_j).$$
Let $(\boldsymbol{\alpha}^*(\hat{x}_a,\hat{x}_b,\mathbf{I}), \boldsymbol{\beta}^*(\hat{x}_a,\hat{x}_b,\mathbf{I}), \boldsymbol{\gamma}^*(\hat{x}_a,\hat{x}_b,\mathbf{I}))$ denote an optimal solution of the dual. A modification of lemma 5 can be used to show that there must exist an optimal solution such that $\boldsymbol{\beta}^*(\hat{x}_a,\hat{x}_b,\mathbf{I})$ is uncrossing.

We consider the values of $\hat{x}_a$ and $\hat{x}_b$ for which we obtain the tight multiplicative bound of the interval move-making algorithm. For these values of $\hat{x}_a$ and $\hat{x}_b$, we define

$$\overline{y}_{ab}(i,j;\hat{x}_a,\hat{x}_b) = \frac{\sum_{\mathbf{I}\in\mathcal{I}}\alpha^*(i,j;\hat{x}_a,\hat{x}_b,\mathbf{I})}{\sum_{i',j'}\sum_{\mathbf{I}\in\mathcal{I}}\alpha^*(i',j';\hat{x}_a,\hat{x}_b,\mathbf{I})}, \forall i,j \in \{1,\cdots,h\},$$

$$\overline{y}_a(i;\hat{x}_a,\hat{x}_b) = \sum_{j=1}^{h}\overline{y}_{ab}(i,j;\hat{x}_a,\hat{x}_b), \forall i \in \{1,\cdots,h\},$$

$$\overline{y}_b(j;\hat{x}_a,\hat{x}_b) = \sum_{i=1}^{h}\overline{y}_{ab}(i,j;\hat{x}_a,\hat{x}_b), \forall j \in \{1,\cdots,h\},$$

$$y'_{ab}(i,j;\hat{x}_a,\hat{x}_b) = \frac{\sum_{\mathbf{I}\in\mathcal{I}}\beta^*(i,j;\hat{x}_a,\hat{x}_b,\mathbf{I})}{\sum_{i',j'}\sum_{\mathbf{I}\in\mathcal{I}}\beta^*(i',j';\hat{x}_a,\hat{x}_b,\mathbf{I})}, \forall i,j \in \{1,\cdots,h\},$$

$$y'_a(i;\hat{x}_a,\hat{x}_b) = \sum_{j=1}^{h}y'_{ab}(i,j;\hat{x}_a,\hat{x}_b), \forall i \in \{1,\cdots,h\},$$

$$y'_b(j;\hat{x}_a,\hat{x}_b) = \sum_{i=1}^{h}y'_{ab}(i,j;\hat{x}_a,\hat{x}_b), \forall j \in \{1,\cdots,h\}.$$

The constraints of the dual of problem (27) ensure that

$$\overline{y}_a(i;\hat{x}_a,\hat{x}_b) = y'_a(i;\hat{x}_a,\hat{x}_b), \forall i \in \{1,\cdots,h\},$$
$$\overline{y}_b(j;\hat{x}_a,\hat{x}_b) = y'_b(j;\hat{x}_a,\hat{x}_b), \forall j \in \{1,\cdots,h\}.$$

Given the distributions $\overline{y}_a(\hat{x}_a,\hat{x}_b)$ and $\overline{y}_b(\hat{x}_a,\hat{x}_b)$, we claim that the pairwise variables $\overline{\mathbf{y}}_{ab}(\hat{x}_a,\hat{x}_b)$ must minimize the pairwise potentials corresponding to these distributions, that is,

$$\overline{\mathbf{y}}_{ab}(\hat{x}_a,\hat{x}_b) = \underset{\mathbf{y}_{ab}}{\operatorname{argmin}} \sum_{l_i,l_j\in\mathbf{L}} d(l_i,l_j)y_{ab}(i,j)$$

$$\text{s.t.} \quad \sum_{l_j\in\mathbf{L}} y_{ab}(i,j) = \overline{y}_a(i;\hat{x}_a,\hat{x}_b), \forall l_i \in \mathbf{L}$$

$$\sum_{l_i\in\mathbf{L}} y_{ab}(i,j) = \overline{y}_b(j;\hat{x}_a,\hat{x}_b), \forall l_j \in \mathbf{L}$$

$$y_{ab}(i,j) \geq 0, \forall l_i, l_j \in \mathbf{L}.$$

If the above claim was not true, then we could further increase the value of at least one of the duals of problem (27) corresponding to some interval $\mathbf{I}$. Furthermore, since $\mathbf{y}'_{ab}(\hat{x}_a,\hat{x}_b)$ is constructed using $\boldsymbol{\beta}^*(\hat{x}_a,\hat{x}_b,\mathbf{I})$, which is uncrossing, a modification of lemma 3 can be used to show that

$$\hat{d}(\hat{\mathbf{y}}_a(\hat{x}_a,\hat{x}_b),\hat{\mathbf{y}}_b(\hat{x}_a,\hat{x}_b)) = \sum_{l_i,l_j\in\mathbf{L}} d(l_i,l_j)y'_{ab}(i,j;\hat{x}_a,\hat{x}_b).$$

Here $\hat{\mathbf{y}}_a(\hat{x}_a,\hat{x}_b)$ and $\hat{\mathbf{y}}_b(\hat{x}_a,\hat{x}_b)$ are the rounded solutions obtained by the interval rounding procedure applied to $\overline{\mathbf{y}}_a(\hat{x}_a,\hat{x}_b)$ and $\overline{\mathbf{y}}_b(\hat{x}_a,\hat{x}_b)$.

By duality, it follows that

$$\frac{\sum_{l_i,l_j\in\mathbf{L}} d(l_i,l_j)y'_{ab}(i,j;\hat{x}_a,\hat{x}_b)}{\sum_{l_i,l_j\in\mathbf{L}} d(l_i,l_j)\overline{y}_{ab}(i,j;\hat{x}_a,\hat{x}_b)} \leq \frac{1}{q} \max_{x_a,x_b,\hat{x}_a,\hat{x}_b\in\mathbf{L},x_a\neq x_b} \frac{D(x_a,x_b;\hat{x}_a,\hat{x}_b)}{d(x_a,x_b)}.$$

Strong duality implies that there must exist a set of variables for which the above inequality holds as an equality. This proves the lemma. $\square$

Lemmas 6 and 8 together prove theorem 2.

## Appendix C: Proof of Theorem 3

*Proof.* The proof of the above theorem proceeds in a similar fashion to the previous two theorems, namely, by computing the dual of the problems that are used to obtain the tightest submodular overestimation of the given distance function. In what follows, we provide a brief sketch of the proof and omit the details.

We start with the hierarchical move-making algorithm. Recall that this algorithm computes a labeling $\mathbf{x}^{i,j}$ for each cluster $\mathbf{C}(i,j) \subseteq \mathbf{L}$, where $\mathbf{C}(i,j)$ denotes the $j$-th cluster at the $i$-th level. In order to obtain a labeling $\mathbf{x}^{i,j}$ it makes use of the labelings computed at the $(i+1)$-th level. Specifically, let $\mathbf{L}_a^{i,j} = \{x_a^{i+1,j'}, p(i+1,j') = j, j' \in \{1, \cdots, h^{i+1}\}\}$. In other words, $\mathbf{L}_a^{i,j}$ is the set of labels that were assigned to the random variable $X_a$ by the labelings corresponding to the children of the current cluster $\mathbf{C}(i,j)$. In order to compute the labeling $\mathbf{x}^{i,j}$, the algorithm has to choose one label from the set $\mathbf{L}_a^{i,j}$. This is achieved either using the complete move-making algorithm or the interval move-making algorithm. The algorithm is initialized by define $x_a^{m,j} = k$ for all $X_a \in \mathbf{X}$ where $k$ is the unique label in the cluster $\mathbf{C}(m,j)$ The algorithm terminates by computing $\mathbf{x}^{1,1}$, which is the final labeling.

Let the multiplicative bound corresponding to the computation of $\mathbf{x}^{i,j}$ be denoted by $B^{i,j}$. From the arguments of theorems 1 and 2, it follows that there must exist dual variables $(\boldsymbol{\alpha}^*(i,j), \boldsymbol{\beta}^*(i,j), \boldsymbol{\gamma}^*(i,j))$ that provide an approximation factor that is exactly equal to $B^{i,j}$ and $\boldsymbol{\beta}^*(i,j)$ is uncrossing. For each cluster $\mathbf{C}(i,j)$, we define variables $\delta_a(k;i,j)$ for $k \in \mathbf{L}_a^{i,j}$ as follows:

$$\delta_a(k;i,j) = \frac{\sum_{k' \in \mathbf{L}_b^{i,j}} \alpha^*(k,k';i,j)}{\sum_{l \in \mathbf{L}_a^{i,j}} \sum_{l' \in \mathbf{L}_b^{i,j}} \alpha^*(l,l';i,j)}.$$

Using the above variables we define unary variables $y_a(k;i,j)$ for each $k \in \mathbf{L}^{i,j}$ as follows:

$$y_a(k;1,1) = \delta_a(k;1,1),$$
$$y_a(k;i,j) = y_a(x_a^{i-1,p(i,j)};i-1,p(i,j))\delta_a(k;i,j).$$

Similarly, we define variables $\delta_b(k';i,j)$ for each $k' \in \mathbf{L}_b^{i,j}$ as follows:

$$\delta_b(k';i,j) = \frac{\sum_{k \in \mathbf{L}_a^{i,j}} \alpha^*(k,k';i,j)}{\sum_{l \in \mathbf{L}_a^{i,j}} \sum_{l' \in \mathbf{L}_b^{i,j}} \alpha^*(l,l';i,j)}.$$

Using the above variables, we define unary variables $y_b(k';i,j)$ for each $k' \in \mathbf{L}_b^{i,j}$ as follows:

$$y_b(k';1,1) = \delta_b(k';1,1),$$
$$y_b(k';i,j) = y_a(x_a^{i-1,p(i,j)};i-1,p(i,j))\delta_b(k';i,j).$$

Since each cluster in the $m$-th level contains a single unique label, it follows that

$$\mathbf{y}_a = [y_a(x_a^{m,j};m,j), \forall j = 1, \cdots, h],$$
$$\mathbf{y}_b = [y_b(x_b^{m,j};m,j), \forall j = 1, \cdots, h],$$

are valid unary variables for the LP relaxation. It can be shown that applying the hierarchical rounding procedure on the variables $\mathbf{y}_a$ and $\mathbf{y}_b$ provides an approximation factor that is exactly equal to the multiplicative bound of the hierarchical move-making algorithm. $\square$