[Reviews · NeurIPS 2014]

Submitted by Assigned_Reviewer_13

This paper focuses on the metric labeling problem, a special case of MAP/MPE inference in graphical models where potentials are restricted to be a metric distance function. It is a theoretical paper that highlights some interesting connections between move-making algorithms (based on a reduction to min-cut) and certain types of rounding schemes used in combination with linear programming relaxations. Specifically, the paper shows that certain move-making algorithms can match the rounding schemes in terms of approximation guarantees, while allowing for the use of fast min cut solvers.

The writeup is excellent. It's a theoretical paper but the authors did a great job making the ideas accessible, highlighting the important concepts without overwhelming the reader with technical details. I like that the paper is mostly self-contained, and overall I found it enjoyable to read.
Although the paper is mainly theoretical, I like that the authors also included an experimental section in the supplementary materials to empirically confirm the theory.

One concern I have is about the novelty of the results. It seems that a number of closely related results were previously known (e.g., reported as corollaries in the supplementary materials). It would be good to specify clearly to what extent Theorems 1, 2 and 3 generalize/extend prior work. Related to this, how do the move-making Algorithms (2, 4 and 6) designed in this paper relate to previous move-making algorithms (e.g., [14,15])? Is there a key difference that allows for a better approximation guarantee?

How expensive is it to solve the linear program (4) needed for move-making? Is it included in the runtime in the experimental results?

Is Algorithm 6 the one used and called HIER in the experimental section in the supplementary materials?

*** EDIT ***
Thanks for the thorough responses - that clears up my concerns.
Summary: It's a well written paper that makes a nice theoretical contribution by providing theoretical guarantees on the approximation obtained with move-making algorithms, a class of algorithms used to solve the metric labeling problem. It studies an important problem in computer vision and the analysis might lead to the development of new inference algorithms for other combinatorial problems.

Submitted by Assigned_Reviewer_33

The authors analyze the relationship between randomized rounding
algorithms and move-making algorithms for metric
labeling, which is the problem of MAP inference in a MRF where the
binary potentials are proportional to a distance metric on the label
space. They demonstrate connections between the two classes of
algorithms. For each of three rounding schemes---complete, interval,
and hierarchical---they present companion move-making algorithms that
achieve the same worst-case approximation ratio (over all possible
unary potentials and edge weights) for any particular distance metric
on the label space.

The primary contribution of the paper is the analysis, which shows
that the move-making algorithms have the same approximation guarantees
as the rounding-based ones. This is significant because the
rounding-based algorithms have historically come with the strongest
approximation guarantees, but move-making algorithms are faster. Thus,
the guarantees are exported to the faster algorithms.

The paper can be situated slightly better in terms of prior work. From
what I can tell:

* Previous papers [14,15] have presented move-making algorithms
similar to those given here and proved approximation factors for
those algorithms for certain classes of distance functions.

* This work, on the other hand, applies to *any* distance function,
and shows that the approximation ratio of the rounding scheme
(for that distance function) is matched by the companion move-making
algorithm. The difference between this and previous results is
subtle and could be emphasized more.

* I believe the move-making algorithms themselves are not
contributions of this paper. It would help a great deal if the
authors could clearly attribute each algorithm to a particular
source and describe which parts are their original contributions.

The paper is well-written and generally well executed. I felt that too
much of the paper was used describing the algorithms in detail
given that the main intellectual contribution seems to be in the
analysis.

A significant question that is left unanswered in the main paper is:
what is the general analysis technique for converting a rounding
scheme to a move-making algorithm? Is there a clear recipe? If not,
what are the significant elements of the analysis? The paper would be
strengthened considerably by providing some discussion of this.
Summary: This is a quality paper that establishes theoretical connections
between LP rounding algorithms for metric labeling and move-making
algorithms. It is well executed, but could do a better job situating
itself with respect to previous work and could provide more insight
into the analysis techniques.

Submitted by Assigned_Reviewer_42

Summary:
This paper addresses the semi-metric labeling problem, a discrete optimization problem whose objective function separates according to a second-order MRF and whose second-order functions are proportional to a semi-metric. It extends recent work [14,15,20] in which (primal) move making algorithms are developed, for some semi-metrics, which match the best known multiplicative bounds of rounded solutions of the LP relaxation. This paper generalizes those results and defines move making algorithms for several rounding procedures and arbitrary semi-metrics.

Quality:
- the paper is technically correct
- the references are adequate
- the authors are clear about the problems they solve and the problems that have been solved before

Clarity:
- the paper is well-written, well-structured and easy to read

Originality:
- the Interval Move Algorithm and the Hierarchical Move Algorithm as well as the equality of the multiplicative bounds of these algorithms to that of related rounding procedures are novel.

Significance:
- this work is a significant contribution to our understanding of the connection between (primal) move making algorithms and algorithms that round the solutions of LP relaxations, in the context of the semi-metric labeling problem.
Summary: A technically correct, well-written and well-structures paper which makes a significant contribution to our understanding of the connection between (primal) move making algorithms and algorithms that round the solutions of LP relaxations, in the context of the semi-metric labeling problem.
Author Feedback
Author rebuttal: We thank the reviewers for carefully reading our paper and providing insightful comments.

Our paper presents an analysis of move-making algorithms for metric labeling that closely mimic rounding procedures used in conjunction with the accurate linear programming (LP) relaxation. Specifically, we show that the multiplicative bounds of the rounding-based move-making algorithms match the approximation factors of the rounding procedures. This allows us to combine the accuracy of the LP relaxation with the efficiency (both in theory and in practice) of move-making algorithms.

All the reviewers found the analysis interesting and technically correct. AR13 and AR42 also commented that the paper is clearly written and well-structured. The main (and valid) criticism is that the paper should make the novelty of the contributions more explicit. To this end, the reviewers have provided specific suggestions to modify the paper, which we will incorporate in subsequent versions. Below, we provide responses to the questions raised in the reviews.

Q: Novelty of algorithms? (AR13 and AR33)

Algorithm 4 (and its special case Algorithm 2) is a minor modification of the method of [15]. [15] considers only truncated convex pairwise potentials, for which there is a readily available submodular overestimation, namely, the corresponding non-truncated convex pairwise potentials. In contrast, Algorithm 4 suggests computing a submodular overestimation for a general semi-metric distance function by solving the small LP (5) (the LP (4) in the case of Algorithm 2).

Algorithm 6 is a minor modification of the method of [14]. While [14] considers the clustering inherently specified by r-HST metrics, Algorithm 6 is applicable for any general clustering.

As suggested by AR33, we will include the above statements in sections 4 and 5 respectively to ensure that the nature of modification is clear to the reader.

Q: Novelty of analysis? (AR13 and AR33)

Our analysis differs significantly from the previous results in two ways. First, the previous results have considered only 4 distance functions (uniform [20], truncated linear [15], truncated quadratic [15] and r-HST [14]), while our analysis is applicable to any semi-metric distance function. Second, we also show the tightness of the theoretical guarantees of the parallel rounding procedures and the move-making algorithms. This removes any possibility of a gap existing between the two seemingly disconnected families of methods for metric labeling.

As noted by AR33, the difference is subtle. We will emphasis it by including the above statements in the introduction and discussion sections of the paper, as well as in the relevant corollaries of the technical report.

Q: Significant elements of the analysis? (AR33)

The most significant element of the analysis is the following: the dual variables of LP (4) (which finds the submodular overestimate of a given distance function) provide a feasible LP solution such that applying complete rounding on this solution has an approximation factor equal to the submodular distortion. So, informally speaking, there is a "duality" between approximation factors of parallel rounding and multiplicative bounds of move-making. This duality is also invariant to applying the rounding on an interval of labels, or over a hierarchical clustering of labels.

Q: General technique for converting rounding procedures to move-making algorithms? (AR33)

The question of whether there is a general technique for converting rounding procedures to move-making algorithms is an important but open one. To the best of our knowledge, there is no move-making algorithm that matches the guarantees of serial rounding [1], so it is possible that such a technique may not exist. Alternately, we can seek a complete characterization of the rounding procedures that can be converted to move-making algorithms. Once again, to the best of our knowledge, this is an open question. We hope that, by proving the existence of move-making algorithms corresponding to the three existing parallel rounding procedures, our paper will bring these questions to the attention of the community. To this end, we will include them in the discussion section of the paper.

Q: Complexity of solving LP (4)? (AR13)

LP (4) has $O(h^2)$ variables and $O(h^2)$ constraints as opposed to the LP relaxation of metric labeling, which has $O(mh^2)$ variables and $O(mh^2)$ constraints. Here, $m$ is the number of edges in the graph corresponding to the MRF (typically of the order of 10^6 for computer vision problems). Hence, solving LP (4) is significantly more efficient than the LP relaxation. Furthermore, LP (4) depends only on the distance function and not the unary potentials and the edge weights. Hence, it needs to be solved only once for a given distance function, after which it can be used to formulate move-making algorithms for any metric labeling problem defined using the given distance function.

Q: Is Algorithm 6 same as HIER? (AR13)

Yes, we will make this clear in the experiments section.